# Test-retest reproducibility of *in vivo* oscillating gradient and microscopic anisotropy diffusion MRI in mice at 9.4 Tesla

Naila Rahman[1,2]*, Kathy Xu[3], Mohammad Omer[1,2], Matthew D. Budde[4], Arthur Brown[3,5], Corey A. Baron[1,2]

**1** Centre for Functional and Metabolic Mapping (CFMM), Robarts Research Institute, University of Western Ontario, London, Ontario, Canada, **2** Department of Medical Biophysics, Schulich School of Medicine and Dentistry, University of Western Ontario, London, Ontario, Canada, **3** Translational Neuroscience Group, Robarts Research Institute, Schulich School of Medicine and Dentistry, University of Western Ontario, London, Ontario, Canada, **4** Department of Neurosurgery, Medical College of Wisconsin, Milwaukee, Wisconsin, United States of America, **5** Department of Anatomy and Cell Biology, University of Western Ontario, London, Ontario, Canada

* nrahma25@uwo.ca

**Data Availability Statement:** All data and analysis files are available in the Open Science Framework (OSF) repository at https://osf.io/5eusw/, DOI: 10.17605/OSF.IO/5EUSW.

## Abstract

### Background and purpose

Microstructure imaging with advanced diffusion MRI (dMRI) techniques have shown increased sensitivity and specificity to microstructural changes in various disease and injury models. Oscillating gradient spin echo (OGSE) dMRI, implemented by varying the oscillating gradient frequency, and microscopic anisotropy (μA) dMRI, implemented via tensor valued diffusion encoding, may provide additional insight by increasing sensitivity to smaller spatial scales and disentangling fiber orientation dispersion from true microstructural changes, respectively. The aims of this study were to characterize the test-retest reproducibility of *in vivo* OGSE and μA dMRI metrics in the mouse brain at 9.4 Tesla and provide estimates of required sample sizes for future investigations.

### Methods

Twelve adult C57Bl/6 mice were scanned twice (5 days apart). Each imaging session consisted of multifrequency OGSE and μA dMRI protocols. Metrics investigated included μA, linear diffusion kurtosis, isotropic diffusion kurtosis, and the diffusion dispersion rate (Λ), which explores the power-law frequency dependence of mean diffusivity. The dMRI metric maps were analyzed with mean region-of-interest (ROI) and whole brain voxel-wise analysis. Bland-Altman plots and coefficients of variation (CV) were used to assess the reproducibility of OGSE and μA metrics. Furthermore, we estimated sample sizes required to detect a variety of effect sizes.

### Results

Bland-Altman plots showed negligible biases between test and retest sessions. ROI-based CVs revealed high reproducibility for most metrics (CVs < 15%). Voxel-wise CV maps

**Funding:** The authors would like to acknowledge the Canada First Research Excellence Fund (BrainsCAN - https://brainscan.uwo.ca/); the New Frontiers in Research Fund (NFRFE-2018-01290 - https://www.sshrc-crsh.gc.ca/funding-financement/nfrf-fnfr/index-eng.aspx), awarded to CB; the Natural Sciences and Engineering Research Council of Canada: Canada Graduate Scholarships - Master's Program (NSERC-CGS M), awarded to NR; and the Ontario Graduate Scholarship (OGS), awarded to NR, for their funding contributions. The funders had no role in study design, data collection and analysis, decision to publish, or preparation of the manuscript.

**Competing interests:** The authors have declared that no competing interests exist.

revealed high reproducibility for µA (CVs ~ 10%), but low reproducibility for OGSE metrics (CVs ~ 50%).

## Conclusion

Most of the µA dMRI metrics are reproducible in both ROI-based and voxel-wise analysis, while the OGSE dMRI metrics are only reproducible in ROI-based analysis. Given feasible sample sizes (10–15), µA metrics and OGSE metrics may provide sensitivity to subtle microstructural changes (4–8%) and moderate changes (> 6%), respectively.

## Introduction

Diffusion MRI (dMRI) provides a non-invasive means to capture microstructure changes in the brain during development, aging, disease, and injury by probing the diffusion of water molecules [1]. The most widely used dMRI techniques are diffusion tensor imaging (DTI) and diffusion kurtosis imaging (DKI). DTI assumes the dMRI signal is entirely characterized by Gaussian diffusion [2] and utilizes a diffusion tensor model to estimate metrics such as mean diffusivity (MD) and fractional anisotropy (FA). DKI provides more information about the underlying tissue via the diffusion kurtosis, which quantifies the deviation from Gaussian diffusion [3]. However, both DTI and DKI are unable to distinguish between microstructural changes and neuron fiber orientation dispersion [2, 4], reducing their specificity to microstructural changes in brain regions with crossing fibers. Furthermore, DKI cannot differentiate between different sources of kurtosis (non-Gaussian diffusion) [3].

Probing microstructure with diffusion-weighted sequences beyond the conventional Stejskal-Tanner pulsed gradient spin echo (PGSE) sequence [5], used in DTI and DKI, is currently of broad interest. The aims of these emerging dMRI sequences are to overcome the limitations of DTI and DKI and improve sensitivity and specificity to microstructural changes. In the present work, the reproducibility of *in vivo* oscillating gradient and microscopic anisotropy dMRI, both of which have unique features that go beyond the PGSE sequence, is investigated in mice at 9.4 Tesla. It is important to note that these are two distinct dMRI methods which are evaluated separately in this work.

The conventional PGSE sequence consists of a pair of pulsed gradients applied along a single direction. Here, the diffusion measurement reflects information about diffusion along a single direction and at a single relatively long diffusion time, which is the time allowed for water molecules to probe the local environment. Given hardware constraints, diffusion times achievable in PGSE can probe displacements on the order of 10–30 µm, which is much larger than typical axon sizes (~ 2 µm) [6].

To overcome the diffusion time limitations of PGSE, the oscillating gradient spin echo (OGSE) method was developed to modify sensitivity to cellular length scales [7]. OGSE allows different microstructure length scales to be probed by varying the frequency of the oscillating diffusion gradients, which is inversely related to diffusion time. For increasing diffusion times (lower oscillating gradient frequencies), the molecules travel greater distances and interact with more barriers such as cell membranes, resulting in lower observed MD values [8]. As MD is different at the various frequencies, this provides the ΔMD—the metric of interest in OGSE dMRI, the difference in MD between the highest and lowest frequencies applied. By acquiring diffusion data at multiple frequencies, the power law relationship between MD and frequency (f) can be explored via the "diffusion dispersion rate", Λ [9, 10]. Evidence of a linear

dependence of MD on the square root of frequency has been demonstrated in healthy and globally ischemic rodent brain tissue [11] and healthy human white matter [12]. Thus, $\Lambda$ can be calculated as

$$\mathrm{MD_f} = \mathrm{MD_0} + \Lambda \cdot \mathrm{f}^{0.5} \tag{1}$$

where $\mathrm{MD_f}$ is the OGSE MD at a frequency f and $\mathrm{MD_0}$ is the MD at f = 0 [9, 10, 12]. Since OGSE is sensitive to structural disorder along one dimension [9], changes in the number and morphology of neurite varicosities will result in changes to $\Lambda$ [10], which potentially makes OGSE an invaluable tool to probe microstructural changes, such as axonal beading, *in vivo* [13, 14].

In contrast to the widely used fractional anisotropy metric (FA), which confounds true microstructural changes with fiber orientation dispersion [2], the microscopic anisotropy (µA) metric quantifies water diffusion anisotropy independent of orientation dispersion [15–17]. To disentangle orientation dispersion from true microstructure changes, the shape of the b-tensor, which describes the strength of diffusion weighting along each direction, is varied via tensor-valued diffusion encoding [16–19]. Most tensor-valued encoding protocols are based on double diffusion encoding (DDE) techniques [16, 20–23] or a combination of linear tensor encoding (LTE) and spherical tensor encoding (STE) [4, 15, 17, 24]. As DDE sequences are implemented via two consecutive diffusion encoding pulses separated by a mixing time, in some cases they may require longer TEs than standard LTE/STE sequences to achieve equal b-values [25]. Conventional DTI and DKI utilize only LTE, in which all gradients are along the same axis, so that diffusion is encoded along a single direction at a time. STE, in which the gradients are distributed throughout all directions, sensitizes the signal to diffusion along all directions at the same time. Here, a combination of LTE and STE is utilized to implement microscopic anisotropy (µA) dMRI [4, 15], using an optimized linear regression technique based on the diffusion kurtosis model [24].

This technique makes the assumption that the dMRI signal arises only from multiple Gaussian components, which may not be appropriate in certain cases, such as when time-dependent diffusion is not negligible [26]. Nevertheless, the normalized signal intensity of powder-averaged dMRI acquisitions of a multi-component system can be represented by the cumulant expansion [17]:

$$\ln\left(\frac{\mathrm{S}}{\mathrm{S_o}}\right) = -\mathrm{bD} + \frac{1}{6}\mathrm{bD^2K}\ldots \tag{2}$$

where S is the powder-averaged signal, $\mathrm{S_o}$ is the mean signal with no diffusion encoding, b is the b-value, D is the diffusivity, and K is the kurtosis of the power-averaged signal. Microscopic anisotropy (µA) is defined here based on the difference in signal between LTE and STE dMRI acquisitions, similar to the equation used in DDE protocols [15, 27]:

$$\mu\mathrm{A} = \sqrt{\frac{\ln\left(\frac{\mathrm{S_{LTE}}}{\mathrm{S_{STE}}}\right)}{\mathrm{b^2}}} \tag{3}$$

where $\mathrm{S_{LTE}}$ and $\mathrm{S_{STE}}$ are the powder-averaged LTE and STE signals, respectively. Microscopic fractional anisotropy (µFA), the normalized counterpart of µA, can be expressed in terms

of µA:

$$\mu FA = \sqrt{\frac{3}{2}\frac{\mu A^2}{\mu A^2 + 0.2D^2}} \tag{4}$$

The complete derivation of Eqs (3) and (4) is available in Arezza et al. [24]. As the LTE signal depends on variance of both isotropic and anisotropic diffusivity, while the STE signal depends only on variance of isotropic diffusivity, diffusional kurtosis estimated from the µA protocol includes linear kurtosis ($K_{LTE}$—arising from the LTE acquisitions) and isotropic kurtosis ($K_{STE}$—arising from the STE acquisitions). $K_{STE}$ is a measure of the variance in the magnitude of diffusion tensors or the mean diffusivity, which can be related to cell size heterogeneity [4].

OGSE and µA dMRI have recently been gaining attention in various disease and injury models and their feasibility has been shown in both preclinical and clinical settings. Importantly, OGSE dMRI can provide measures of mean cell size [28, 29] and axonal diameter [30–33], while µA dMRI can provide estimates of cell shape [4, 16–20, 22, 23]. Low-frequency OGSE has also been shown to provide better contrast, compared to PGSE, to cylinder diameter in the presence of orientation dispersion [31, 32, 34]. The OGSE ΔMD metric has shown increased sensitivity, compared to MD alone, in the assessment of hypoxia-ischemia [35] and radiation therapy treatment response [36] in rodents, and in various pathologies in humans, including muscle contraction abnormalities [37], high- and low-grade brain tumor differentiation [38], and neonatal hypoxic-ischemic encephalopathy [39]. Notably, OGSE has helped to identify neurite beading as a mechanism for dMRI contrast after ischemic stroke [13, 14]. Preliminary studies in humans have found that µA provides better sensitivity than the conventional FA in distinguishing between different types of brain tumours [4], assessment of multiple sclerosis lesions [38, 40], and detecting white matter microstructure changes associated with HIV infection [27]. Furthermore, Westin et al. reported that $K_{STE}$ showed significant differences between controls and schizophrenia patients, while conventional mean kurtosis showed no difference [41]. The feasibility of µA dMRI has been demonstrated in rodents both *in vivo* [42, 43] and *ex vivo* [26, 27, 44]. *In vivo* preclinical rodent µA studies, which have included predominantly DDE techniques and more recently combined LTE/STE techniques, have shown that measurements of eccentricity provide additional sources of contrast for the rat brain, especially in the gray matter [42], and recently, He et al. have shown that $K_{STE}$ may be particularly sensitive to deep gray matter lesions [45].

As dMRI has reached the forefront of tissue microstructure imaging [46], there is a need to establish the reproducibility of these emerging methods. While the reproducibility of DTI and DKI has been investigated extensively [47–50], to the best of our knowledge, no test-retest assessment of OGSE and µA dMRI has been done at ultra-high field strength. The aim of this work was to assess test-retest reproducibility of *in vivo* OGSE and µA dMRI in mice at 9.4 Tesla and provide estimates of required sample sizes, which is essential in planning future preclinical neuroimaging studies involving models of disease/injury.

## Methods

### Subjects

All animal procedures were approved by the University of Western Ontario Animal Use Subcommittee and were consistent with guidelines established by the Canadian Council on Animal Care. Twelve adult C57Bl/6 mice (six male and six female), between 12–14 weeks old, were scanned twice 5 days apart. The sample size was chosen to reflect similar sample sizes

used in other pre-clinical imaging studies [51–54]. Before scanning, anesthesia was induced by placing the animals in an induction chamber with 4% isoflurane and an oxygen flow rate of 1.5 L/min. Following induction, isoflurane was maintained during the imaging session at 1.8% with an oxygen flow rate of 1.5 L/min through a custom-built nose cone. The mouse head was fixed in place using ear bars and a bite bar to prevent head motion. These mice were part of a longitudinal study, at the end of which they were euthanized for histology. The mice were anesthetized with ketamine/xylazine (2:1) and then underwent trans-cardiac perfusion with ice-cold saline, followed by 4% paraformaldehyde in phosphate-buffer saline (PBS).

### *In vivo* MRI

*In vivo* MRI experiments were performed on a 9.4 Tesla (T) Bruker small animal scanner equipped with a gradient coil set of 1 T/m strength (slew rate = 4100 T/m/s). A single channel transceive surface coil (20 mm x 25 mm), built in-house, was fixed in place directly above the mouse head to maximize signal-to-noise ratio (SNR). The mouse holder (which included the ear bars and bite bar), nose cone, and surface coil were fixed onto a support, which was placed into the scanner. This ensured consistent positioning of the mouse head in the scanner at each session. Each dMRI protocol was acquired with single-shot spin echo echo-planar-imaging (EPI) readout with scan parameters: TR = 10 s; in-plane resolution = 175 x 200 μm; slice thickness = 500 μm; 30 slices to acquire the full brain; field-of-view = 19.2 x 14.4 mm$^2$; partial Fourier imaging in the phase encode direction with 80% of k-space being sampled; 45 minutes scan time. For each dMRI protocol, a single reverse phase encoded b = 0 s/mm$^2$ volume was acquired at the end of the diffusion sequence for subsequent use in TOPUP [55] and EDDY [56] to correct for susceptibility and eddy current induced distortions. Averages were acquired separately on the scanner and combined using in-house MATLAB code which included reconstruction of partial Fourier data using POCS (Projection onto Convex Sets) [57] and correction for frequency and signal drift associated with gradient coil heating [58]. Anatomical images were also acquired for each subject within each session using a 2D T2-weighted Tur-boRARE pulse sequence (150 μm in-plane resolution; 500 μm slice thickness; TE/TR = 40/5000 ms; 16 averages; total acquisition time = 22 min).

**Oscillating gradient spin echo (OGSE) dMRI.** The OGSE dMRI protocol included a PGSE sequence (with gradient duration = 11 ms and diffusion time = 13.8 ms) and four OGSE sequences with oscillating gradient frequencies of 50 Hz, 100 Hz, 145 Hz, and 190 Hz. The waveforms and gradient modulation power spectra are shown in Fig (1A)–(1E). The 50 Hz sequence is based on the recently proposed frequency tuned bipolar (FTB) oscillating gradient waveform, which allows for shorter TEs at lower frequencies [59]. The frequencies were chosen based on a hypoxic-ischemic injury study in mice [35], where the frequencies ranged from 0–200 Hz, which enables probing length scales between 1.2–4.2 μm. Other scan parameters included: gradient separation = 5.5 ms; TE = 39.2 ms; 5 averages; b = 800 s/mm$^2$; 10 diffusion encoding directions. As the gradient duration was slightly different for each OGSE sequence, zeroes were added to the start of the first diffusion-encoding waveform and to the end of the second diffusion-encoding waveform, to ensure that TE remained the same across all OGSE sequences. 10 b = 0 s/mm$^2$ volumes were interspersed evenly throughout the acquisition.

**Microscopic anisotropy (μA) dMRI.** The STE dMRI gradient waveforms implemented here were similar to the protocol in Arezza et al. [24]. The μA sequence was implemented with linear (LTE) and spherical tensor (STE) encodings, as shown in Fig (1F) and (1G), at b = 2000 s/mm$^2$ (30 directions for each of LTE and STE) and b = 1000 s/mm$^2$ (12 directions). Other scan parameters included: gradient duration = 5 ms; gradient separation = 5.54 ms; TE = 26.8 ms; 3 averages. 8 b = 0 s/mm$^2$ volumes were interspersed evenly throughout the acquisition.

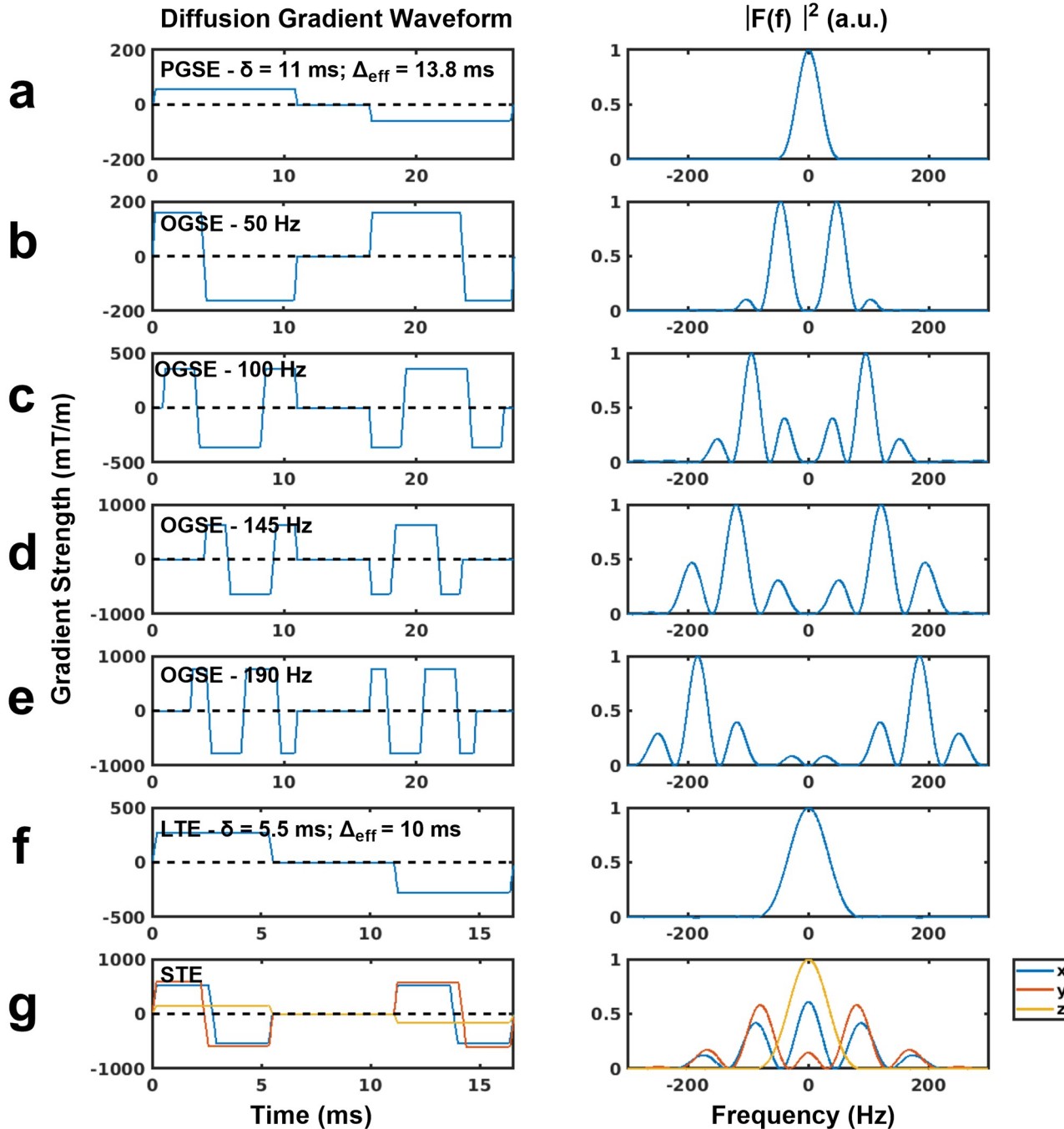

**Fig 1.** Schematic representations of the gradient waveforms and gradient modulation power spectra, $|F(f)|^2$, used for the PGSE (a), OGSE (b-e) and μA (f-g) protocols. Diffusion encoding blocks have been inserted on both sides of a 180° pulse and implicit gradient reversal due to the 180° pulse has been applied. The PGSE waveform (a) is shown with a gradient duration of 11 ms and diffusion time of 13.8 ms. OGSE waveforms (b-e) with gradient oscillation frequencies of 50 Hz, 100 Hz, 145 Hz, and 190 Hz have zeros added to the start of the first gradient and end of the second gradient to ensure all scans in the OGSE protocol have the same TE. LTE and STE waveforms, used in the μA protocol, are shown in (f) and (g) respectively.

## Image processing

Images were pre-processed using PCA denoising [60] and Gibbs ringing correction from the MRtrix3 package [61], followed by TOPUP [55] and EDDY [56] from FMRIB Software Library

(FSL, Oxford, UK) [62]. Brain masks were produced using the skull stripping tool from Brain-Suite (v. 19b) [63]. Image registration was performed using affine and symmetric diffeomorphic transforms with ANTs software (https://github.com/ANTsX/ANTs) [64]. Region-of-interest (ROI) masks were acquired from the labeled Allen Mouse Brain Atlas [65]. Since registration to an atlas is time-consuming, only one anatomical T2-weighted scan was chosen (the "chosen T2") to be registered to the atlas. All other anatomical T2-weighted images were registered to the chosen T2. Non-diffusion weighted (b0) volumes were registered to the corresponding anatomical images (from the same subject at the same timepoint). All dMRI volumes were registered to the corresponding anatomical space using the transforms resulting from the previous step (b0 → corresponding T2). For ROI-based analysis, the inverse transforms resulting from these two registration steps (corresponding T2 → chosen T2 → atlas) were then used to bring the labeled atlas to the corresponding T2 space for each subject at each timepoint. Binary masks for each ROI were generated by thresholding the labeled atlas. Each mask was eroded by one voxel, except for the corpus callosum masks, to minimize partial volume errors within a given ROI. The binary masks were visually inspected to ensure good registration quality. Furthermore, to perform whole brain voxel-wise analysis of all subjects across both timepoints, all dMRI volumes were registered to the chosen T2 space using transforms from two registration steps (b0 → corresponding T2 → chosen T2). For voxel-wise analysis targeted to specific ROIs, the labeled atlas was registered to the chosen T2 space.

From the OGSE data, maps of MD at each frequency were generated using MRtrix3 [61, 66]. ΔMD was calculated as the difference between MD acquired at the highest frequency (190 Hz) and MD acquired at the lowest frequency (0 Hz). To characterize the power law relationship between MD and OGSE frequency (f) [10], the slope of linear regression of MD with $f^{0.5}$, the diffusion dispersion rate (Λ), was calculated. From the μA data, maps of μA, μFA, $K_{LTE}$, and $K_{STE}$ were generated by fitting the powder-averaged STE and LTE signals versus b-value to the diffusion kurtosis model, using a joint non-negative least squares method assuming consistent diffusivity between STE and LTE [24]. As a reference for the OGSE and μA metrics, DTI metrics (MD and FA) have been included in both ROI-based and voxel-wise variability analyses. MD and FA maps were generated using the PGSE sequence (with b = 800 s/mm$^2$ from the OGSE protocol) and the LTE sequence (with b = 1000 s/mm$^2$ from the μA protocol), separately.

## Data and sequence availability

The test-retest dataset and pulse sequences used in this work are available online [67, 68].

## Data analysis

Measurement reproducibility was explored for both ROI-based analysis and whole brain voxel-wise analysis, since both are common analyses techniques in neuroimaging. To mitigate partial volume errors from cerebrospinal fluid (CSF), voxels with MD (0 Hz) > 0.9 μm$^2$/ms were omitted from the analyses of all scalar maps. Outlier detection was included in both ROI-based and voxel-wise analyses, to remove data both animal-wise and voxel-wise. Outliers were defined as values which were more than three scaled median absolute deviations (MAD) away from the median. The ROI analysis focused on five different tissue regions: corpus callosum, internal capsule, hippocampus, cortex, and thalamus. Bland-Altman analysis was performed for both ROI-based and voxel-wise analyses to identify any biases between test and retest measurements. For both analysis techniques, the scan-rescan reproducibility was characterized using the coefficient of variation (CV). The CV reflects both the reproducibility and variability of these metrics and allows calculation of the sample sizes necessary to detect various effect

sizes. CVs were calculated between subjects and within subjects to quantify the between subject and within subject reproducibility respectively. The between subject CV was calculated separately for the test and retest timepoints as the standard deviation divided by the mean value across subjects 1–12. These two CV values were then averaged for the mean between subject CV. The within subject CV was calculated separately for each subject as the standard deviation divided by the mean of the test and retest scans. The 12 within subject CVs were then averaged to determine the mean within subject CV. Following the procedure presented in van Belle [69], the between subject CVs, from the ROI analysis, were used to determine the sample size required per group to detect a defined biological effect between subjects in each ROI. Assuming paired t-tests, the standard deviations of the differences between test-retest mean values across subjects, were used to determine the sample size required to detect a defined biological effect within subjects in each ROI [70]. The minimum sample sizes, using the between and within subject approaches, were both determined at a 95% significance level ($\alpha = 0.05$) and power of 80% ($1-\beta = 0.80$).

**SNR analysis.** As the transceive surface coil used in this study was built in-house, SNR maps were generated for the lowest and highest b-value acquisitions in the OGSE and µA protocols to compare SNR acquired using a commercially available 40-mm millipede (MP40) volume coil (Agilent, Palo Alto, CA, USA) and SNR acquired with fewer averages. SNR maps were calculated by dividing the powder-averaged magnitude signal (of the combined averages) by the noise. Noise was calculated from each of the real and imaginary components of the complex-valued data as the standard deviation of the background signal from a single average of a single direction divided by $\sqrt{(\text{number of averages}) \cdot (\text{number of directions})}$, and averaged over the real and imaginary components. Furthermore, to test the effects of using a different number of averages on the results, ROI-based between and within-subject CV analysis was performed on subsets of the OGSE and µA data containing only 3 and 2 averages, respectively. Note that preprocessing was performed on the subset of fewer averages separately from the full data set (e.g., denoising only used the subset of averages).

**ROI analysis.** The mean MD was calculated for each ROI at each frequency. For each ROI, ΔMD was calculated as the difference between the mean MD at 190 Hz and the mean MD at 0 Hz. The apparent diffusion dispersion rate, Λ, was determined for each ROI by performing a least square fit of the mean MD (in each ROI) to $f^{0.5}$. Scalar maps from the µA protocol ($\mu A$, $\mu FA$, $K_{LTE}$, $K_{STE}$) were computed directly from the signal, and mean values for each metric were computed for each ROI. It should be noted that for both OGSE and µA metrics, averaging for each ROI was performed over the first non-signal parameter computed. Bland-Altman and CV analyses were performed using the mean values.

**Voxel-wise analysis.** ΔMD maps were generated by subtracting the MD maps at 0 Hz from the MD maps at 190 Hz. Λ maps were generated by performing a least square fit of MD to $f^{0.5}$ for each voxel. Voxel-wise Bland-Altman and CV analyses were performed for each metric using the scalar maps (ΔMD, Λ, and scalar maps from the µA protocol).

## Results

### SNR analysis

SNR maps, shown in Fig 2, revealed a higher SNR in the cortex when using the surface coil (with 5 and 3 averages for the OGSE and µA protocols respectively) compared to the MP40 volume coil. As expected, a gradient of SNR change can be seen in the superior-inferior direction of the brain with the surface coil.

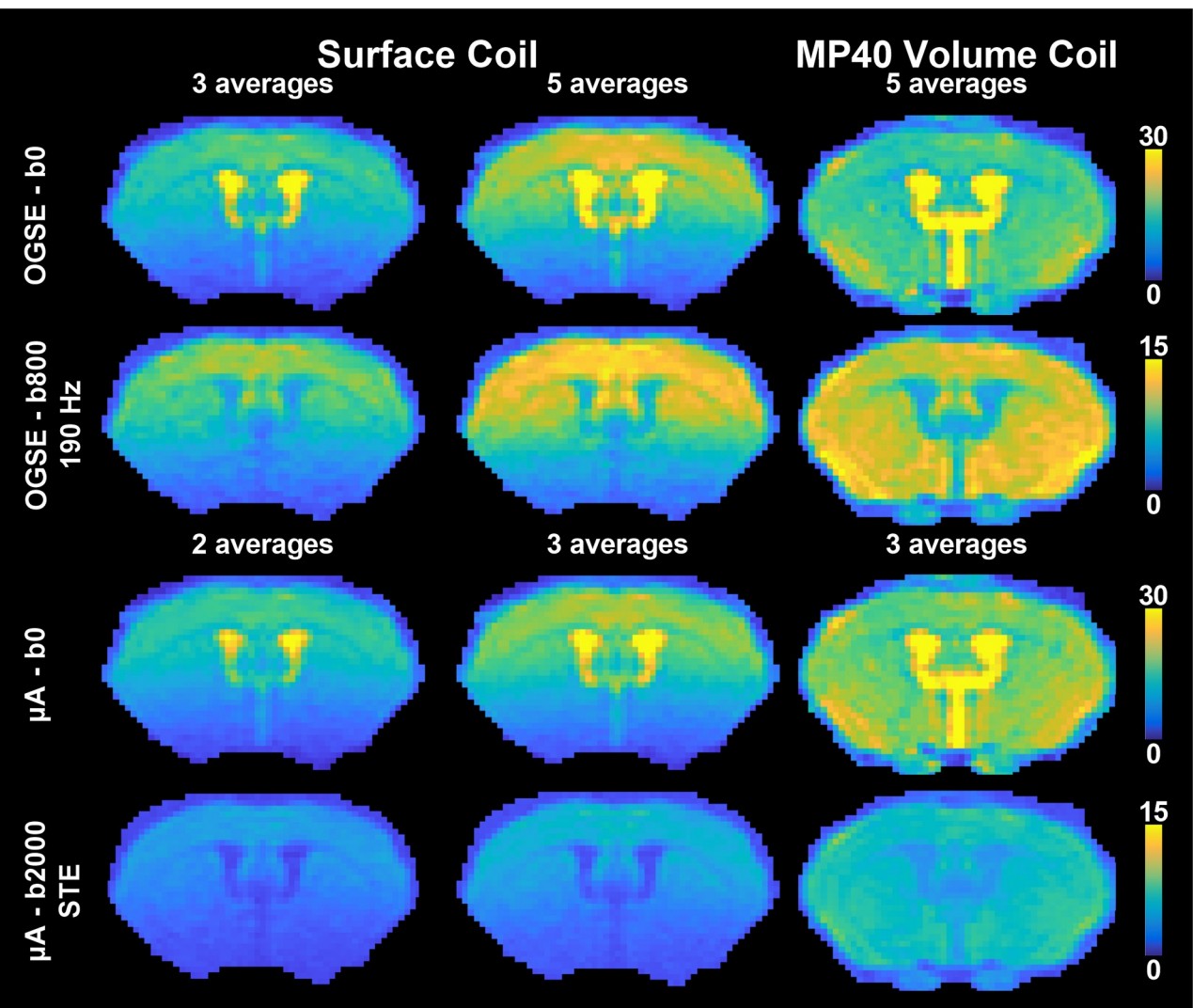

**Fig 2. SNR maps of images acquired with the surface coil and the millipede-40mm (MP40) volume coil.** SNR maps for a single b = 0 s/mm² image are shown for both protocols, and SNR maps for the powder average of the highest b-values are shown for both protocols (b = 800 s/mm² for OGSE-190 Hz and b = 2000 s/mm² for μA-STE). The middle column shows SNR maps acquired from the surface coil with the number of averages used in this study (5 averages for the OGSE protocol and 3 averages for the μA protocol). The left column shows the effect of using fewer averages (3 averages for OGSE and 2 averages for μA). The right column shows the effect of using a commercially available MP40 volume coil with same number of averages used in this study.

### Raw data to parameter maps

Raw data (after combining all averages) and preprocessed data are shown in Fig 3. Representative parameter maps are shown in Fig 4. MD (190 Hz) has an overall higher intensity than MD (0 Hz). ΔMD shows selective enhancement of distinct regions in the brain—the dentate gyrus (part of the hippocampal formation) is shown with white arrows. As expected, ΔMD and Λ show similar contrast. ROI-based fitting of Λ showed the expected trends with $f^{0.5}$ in all ROIs and at both test and retest time-points (Fig 5). The μA and μFA maps also show similar contrast. $K_{LTE}$ highlights white matter structures as expected and $K_{STE}$ is homogenous throughout the brain, although very high in CSF regions and regions impacted by CSF partial volume effects.

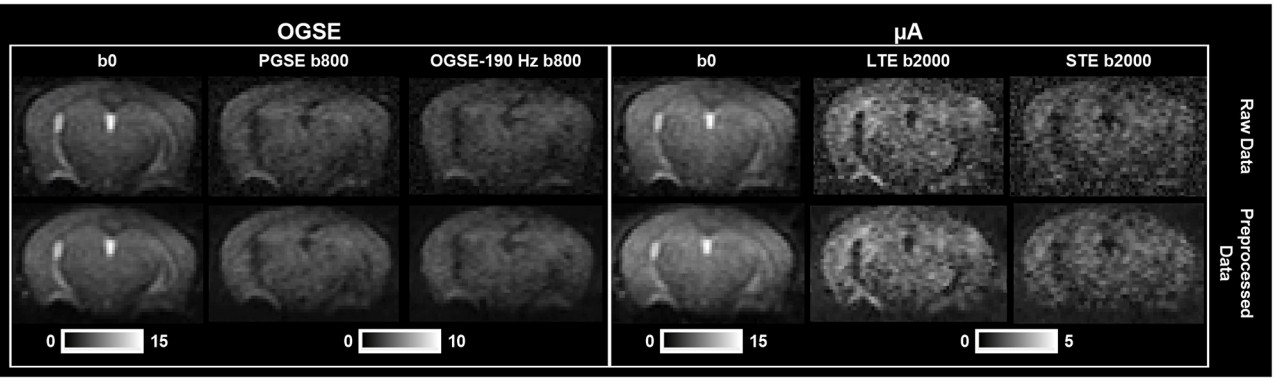

**Fig 3. Raw and preprocessed dMRI data.** Raw data (after combining averages) is shown in the top row and preprocessed data is shown in the bottom row. Representative b = 0 s/mm² images are shown for both the OGSE and µA protocols. From the OGSE protocol, representative diffusion weighted images from a single diffusion gradient direction are shown from PGSE and OGSE with the highest frequency used in this study (190 Hz), at b = 800 s/mm². From the µA protocol, diffusion weighted images from a single diffusion gradient direction are shown from the LTE and STE sequences, at b = 2000 s/mm².

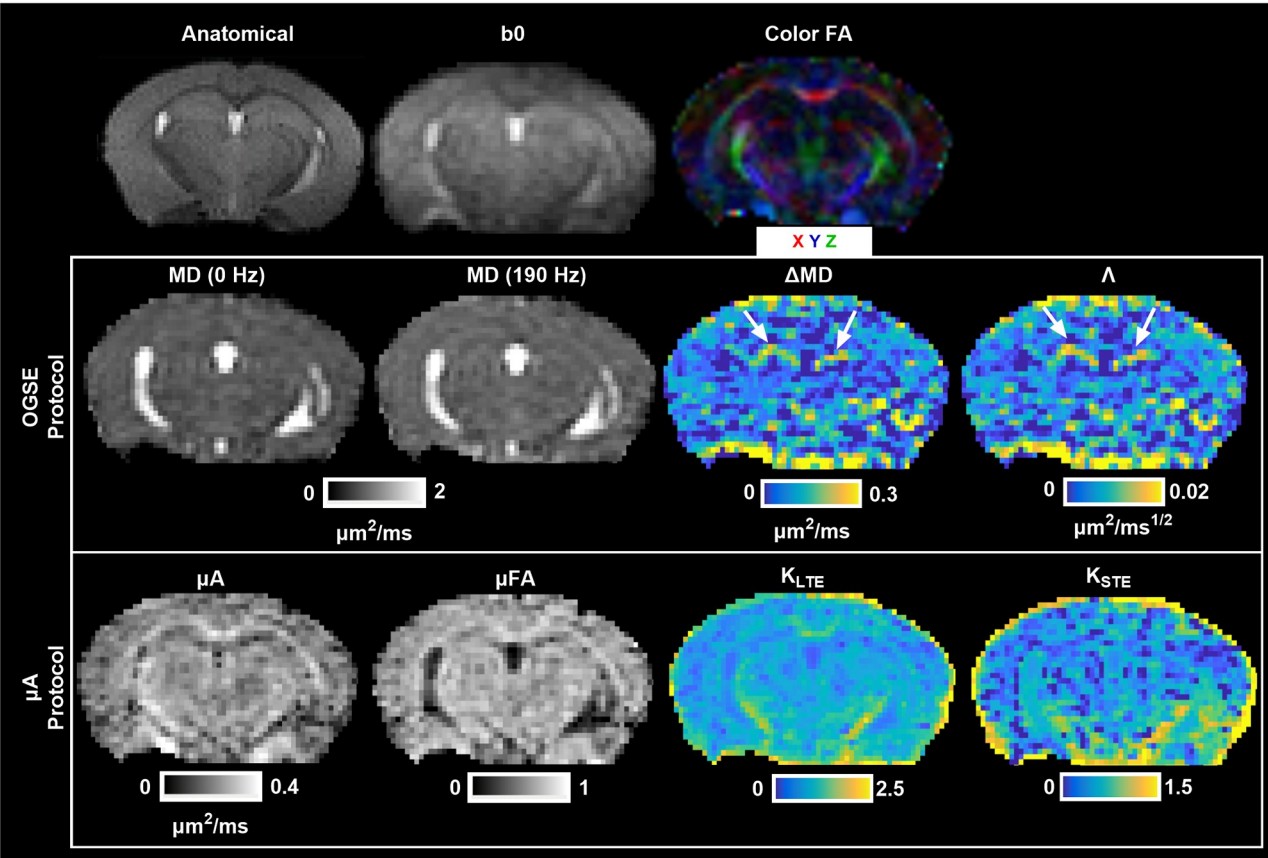

**Fig 4. Example axial cross sections from a single subject showing an anatomical T2-weighted image, a non-diffusion weighted image (b0), and a color fractional anisotropy map (Color FA), where the colors represent the primary direction of diffusion.** Parameter maps from the OGSE protocol (MD (0 Hz): Mean Diffusivity from PGSE (0 Hz); MD (190 Hz): Mean Diffusivity from OGSE (190 Hz); ΔMD: the difference between MD (190 Hz) and MD (0 Hz); Λ: the apparent diffusion dispersion rate) and the µA protocol (µA: Microscopic Anisotropy; µFA: Microscopic Fractional Anisotropy; $K_{LTE}$: Linear Kurtosis (from linear tensor encodings); $K_{STE}$: Isotropic Kurtosis (from spherical tensor encodings)) are shown. The white arrows in the ΔMD and Λ maps indicate high OGSE contrast in the dentate gyrus.

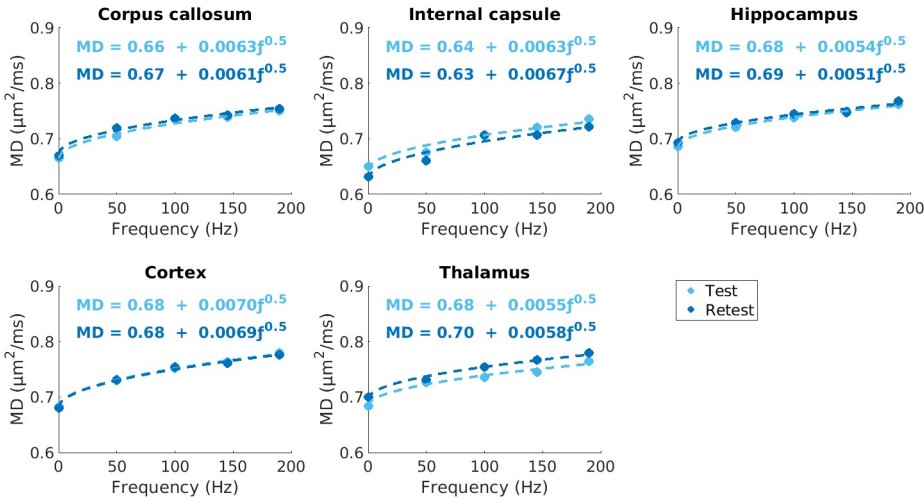

**Fig 5. Least square fitting of mean MD values to $f^{0.5}$, depicted by the dotted lines, in each ROI for test and retest timepoints in one mouse.** The diffusion dispersion rate, $\Lambda$, ranged from 0.0051–0.0070 $\mu m^2/ms^{1/2}$, depending on the ROI.

## ROI analysis

Violin plots depict the distribution of the mean values for each metric within each ROI for the twelve subjects (Fig 6). Across all metrics, the median and interquartile range are similar for test and retest conditions. In general, the smaller ROIs (the internal capsule and the thalamus) show greater distributions, while the larger ROIs (i.e., the cortex) showed much tighter distributions. Bland-Altman plots (Fig 7) revealed negligible biases between repeat measurements across all metrics. In the variability analysis, outlier removal revealed at most one outlier for most metrics (in some of the ROIs), with $K_{STE}$ and FA containing two outliers in the thalamus and internal capsule. $\Lambda$ and $\Delta MD$ showed similar CVs, with the between subject CVs ranging from 5–15%, and the within subject CVs ranging from 4–10%. (Fig 8). µA and µFA show low between and within subject CVs for all ROIs (ranging from 3–8%), with µFA showing slightly lower CVs. $K_{LTE}$ exhibited consistently lower between and within subject CVs (3–8%) compared to $K_{STE}$ (10–17%). In terms of the DTI metrics, the lowest CVs were observed in MD (CVs < 5%) and FA showed a higher variation of CVs than most of the OGSE and µA metrics. ROI-based between and within-subject CV analysis performed on OGSE and µA data with fewer averages revealed comparable CVs (as shown in S1 Fig).

## Voxel-wise analysis

Bland-Altman plots comparing whole brain test and retest voxels for all twelve subjects revealed negligible biases for all metrics (Fig 9). However, $\Delta MD$, $\Lambda$, and $K_{STE}$ showed greater variation in test and retest differences. The CV maps (Fig 10) show very high CVs in the CSF regions (except for the $K_{STE}$ and FA CV maps). Histograms (Fig 11) show $\Delta MD$ and $\Lambda$ have the same distribution. Overall, the between and within subject CVs are comparable for all metrics. µA, µFA, and $K_{LTE}$ have comparable CVs with peaks at 10, 8, and 16% respectively. $\Delta MD$, $\Lambda$, and $K_{STE}$ peak around 50% and have very wide distributions. In comparison, the DTI metrics, MD and FA, peak at 8% and 25% respectively. Whole brain histograms and histograms for specific ROIs (S2 Fig) show similar trends.

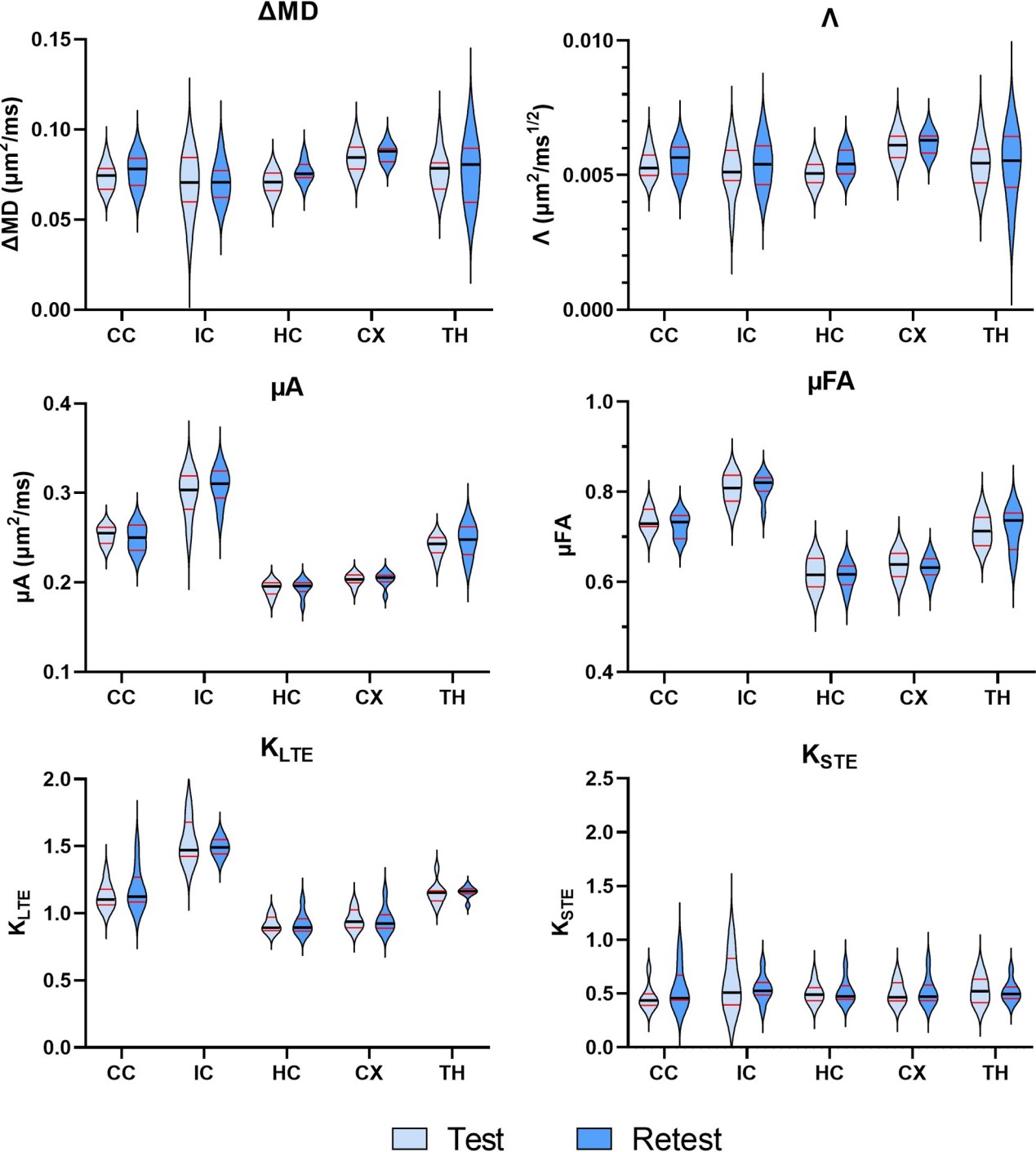

**Fig 6. Violin plots showing the distribution of the OGSE metrics (ΔMD and Λ) and the µA metrics (µA, µFA, $K_{LTE}$, and $K_{STE}$) at the test and retest timepoints (five days apart) for twelve subjects in several brain regions.** The dark black line represents the median and the red lines depict the interquartile range (25th to 75th percentile). The violin plots extend to the minimum and maximum values of each metric. ROIs are abbreviated as follows: CC—corpus callosum; IC—internal capsule; HC—hippocampus; CX—cortex; TH—thalamus.

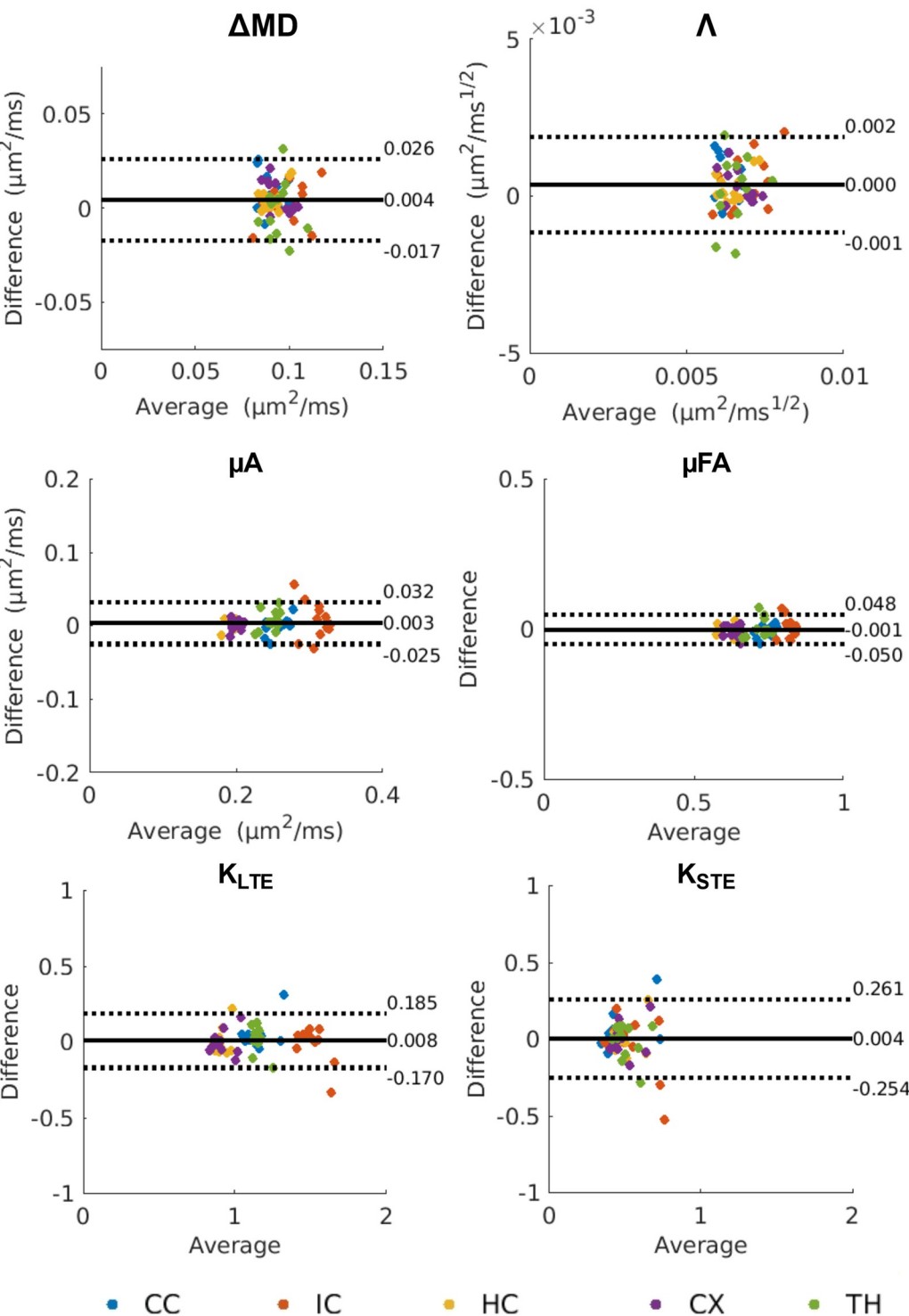

**Fig 7. Bland-Altman plots depicting biases between test and retest scans for mean values of OGSE and µA metrics (from the ROI-based analysis).** The solid black lines represent the mean bias, and the dotted black lines represent the ±1.96 standard deviation lines. The average of the test and retest mean values is plotted along the x-axis and the difference between the test and retest mean values is plotted along the y-axis. ROIs in the legend are abbreviated as follows: CC—corpus callosum; IC—internal capsule; HC—hippocampus; CX—cortex; TH—thalamus.

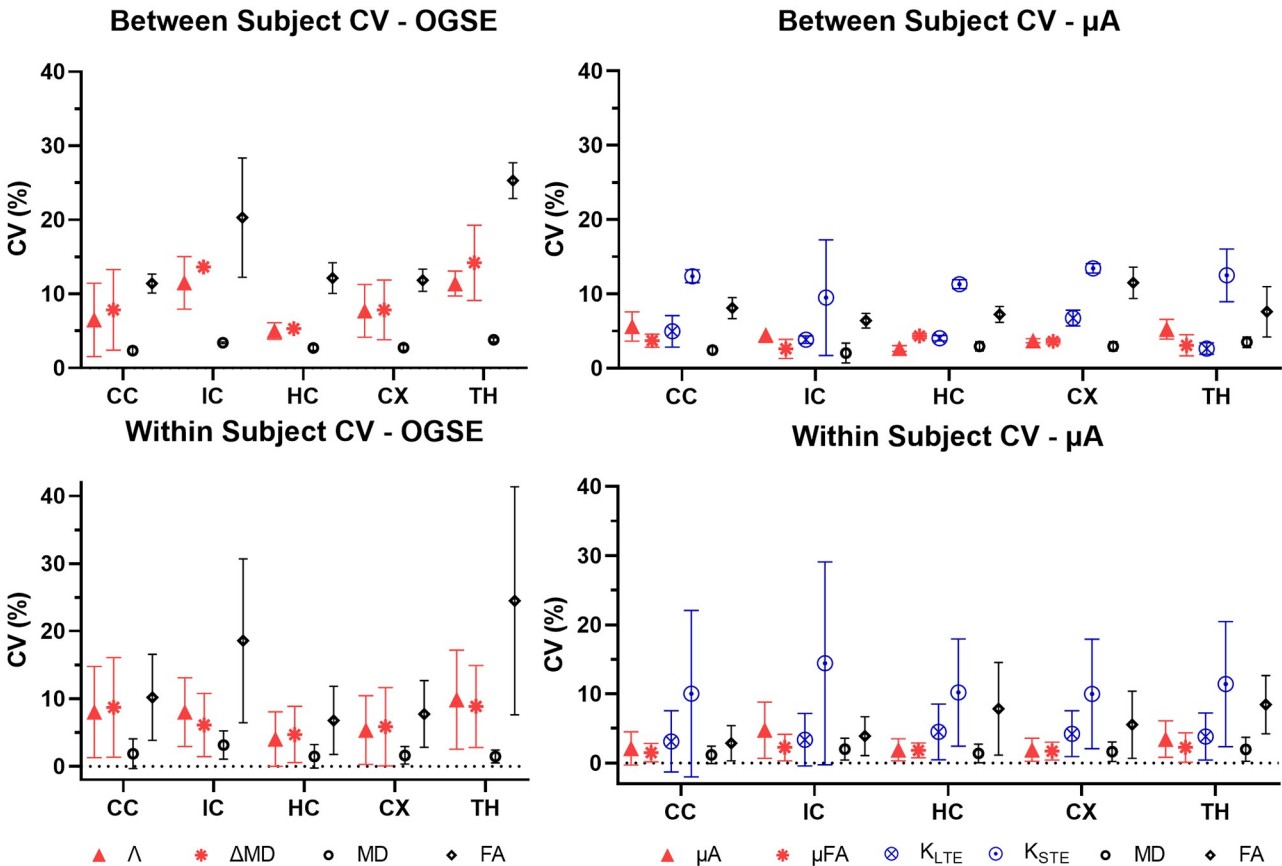

**Fig 8. Mean between subject and within subject coefficients of variation (CV) for OGSE and μA metrics for each ROI.** Values for the between subject condition represent the mean ± standard deviation over subjects (averaged over the test and retest timepoints). Values for the within subject condition represent the mean ± standard deviation between test and retest (averaged over all subjects). ROIs are abbreviated as follows: CC—corpus callosum; IC—internal capsule; HC—hippocampus; CX—cortex; TH—thalamus.

### Sample sizes and minimum detectable effect

**Between subjects.** The between subject CVs, from the ROI analysis, were used to determine the minimum sample sizes required to detect statistically significant changes of 4, 6, 8, 10, and 12% between subjects in each metric within each ROI. ΔMD required a sample size of 15 to detect a minimum change of 8% in the three larger ROIs (the corpus callosum, hippocampus, and cortex). In comparison, the same changes could be detected in Λ with a sample size of 9 (Fig 12). μA and μFA required a sample size of 9 to detect a 6% change in the three larger ROIs. With a sample size of 12, a minimum change of 8% in $K_{LTE}$ could be detected within all ROIs. $K_{STE}$, on the other hand, required much larger sample sizes (at least 20 subjects per group are required to detect a 12% change in the three larger ROIs).

**Within subjects.** The standard deviations of the differences between test-retest mean values across subjects (assuming paired t-tests) were used to determine the minimum sample sizes required to detect statistically significant changes of 4, 6, 8, 10, and 12% within subjects in each metric within each ROI. In the larger ROIs, changes on the order of 8–10% could be detected in Λ with 12 subjects per group, while ΔMD showed similar trends, requiring 15 subjects per group to detect changes of 8–10% (Fig 13). μA was able to detect a minimum change of 4% in the larger ROIs with 12 subjects per group, while the smaller ROIs required greater

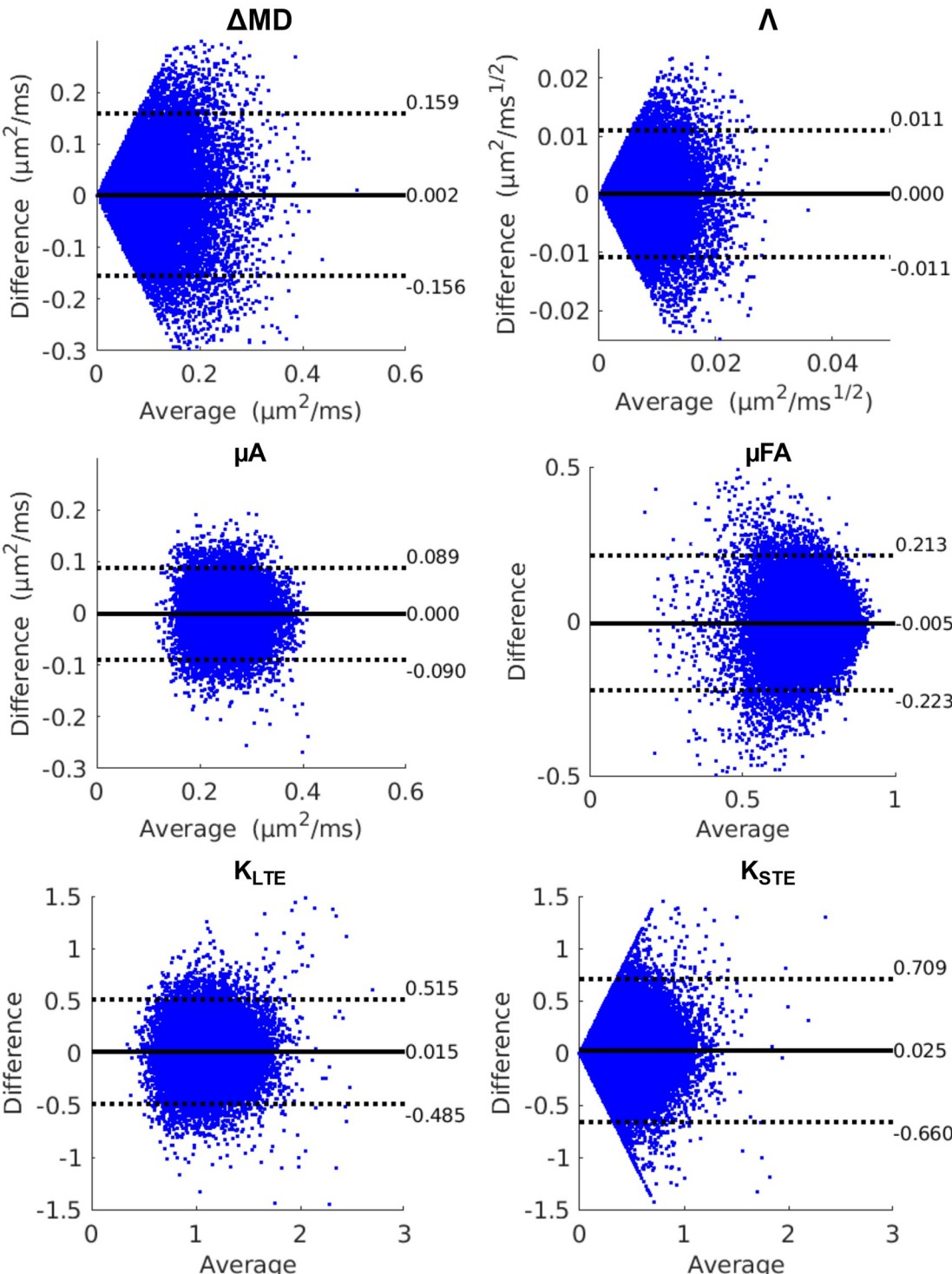

**Fig 9. Bland-Altman plots depicting biases between test and retest scans for OGSE and µA metrics from the whole-brain voxelwise analysis for all subjects.** The solid black lines represent the mean bias, and the dotted black lines represent the ±1.96 standard deviation lines. The average of the test and retest voxels is plotted along the x-axis and the difference between the test and retest voxels is plotted along the y-axis.

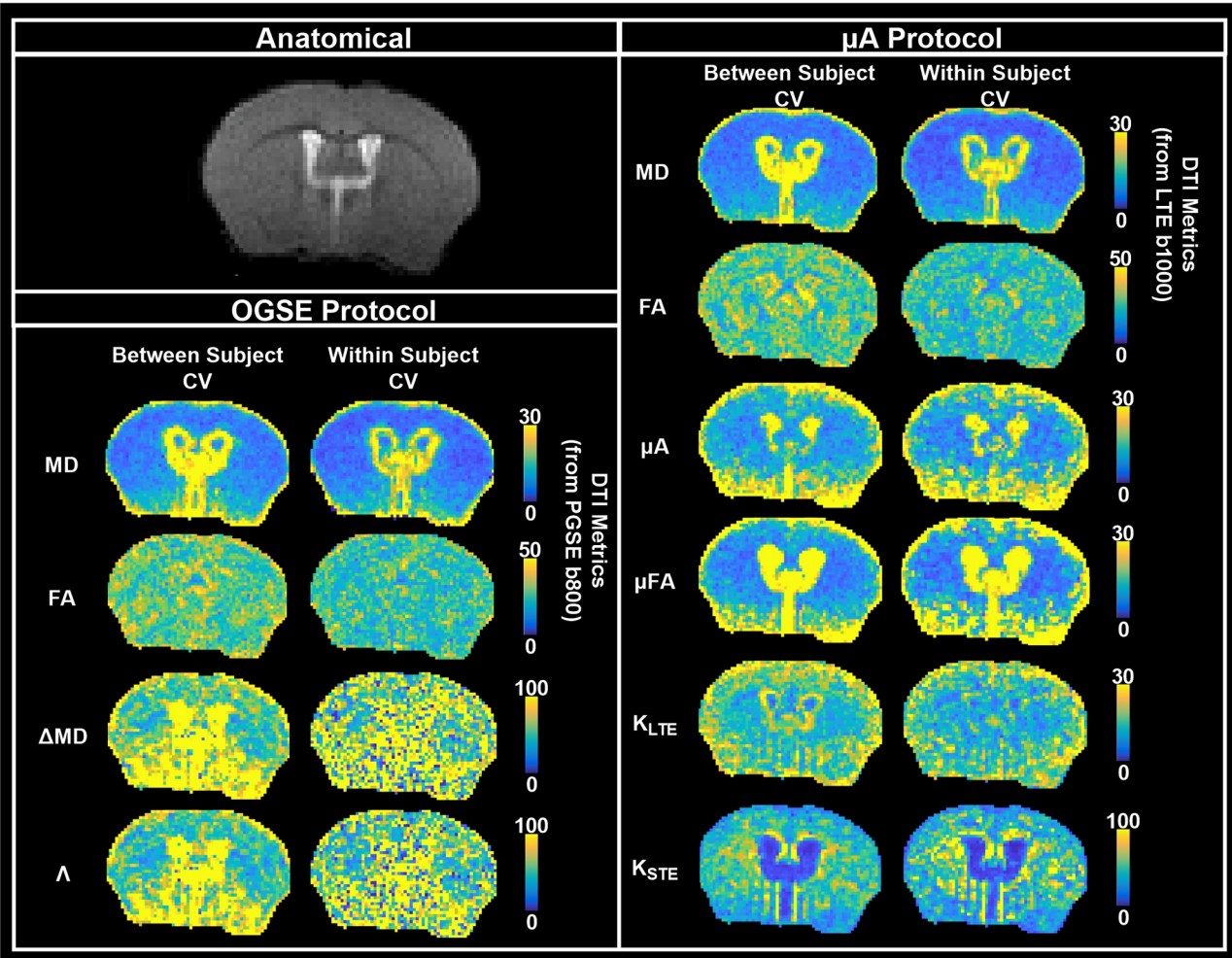

**Fig 10. Whole brain average between subject and within subject CV maps.** All diffusion data was registered to a single anatomical T2-weighted dataset (representative axial slice shown). Values for the between subject condition represent the mean CV within each voxel averaged over the test and retest timepoints. Values for the within subject condition represent the mean CV within each voxel averaged over all subjects. DTI metrics, MD and FA, acquired from both the OGSE and μA protocols, are shown as a reference. Note that the color bar scale varies between the maps.

sample sizes. μFA was slightly more robust, being able to detect a 4% change in the larger ROIs (with 9 subjects per group) and in all ROIs (with 14 subjects per group). $K_{LTE}$ was able to detect moderate changes (6%) with 12 subjects per group in all ROIs, whereas $K_{STE}$ required at least 30 subjects to detect larger changes (12%).

## Discussion

This study explored the reproducibility of OGSE and μA metrics at 9.4 Tesla. No biases were found between repeat measurements with either ROI-based or voxel-wise analysis. μA, μFA, and $K_{LTE}$ were shown to be reproducible in both the mean ROI analysis and the whole brain voxel-wise analysis, while ΔMD, Λ, and $K_{STE}$ were reproducible in only the mean ROI analysis. μA and μFA showed the highest reproducibility of all the metrics, comparable to the DTI metric MD, and the least dispersion of CVs. The CVs observed for μFA in this work are consistent with CVs reported in a recent study by Arezza et al. [24] in human subjects at 3 T, where CVs ranged from 6–8%. Overall, within subject CVs were lower than between subject CVs for both

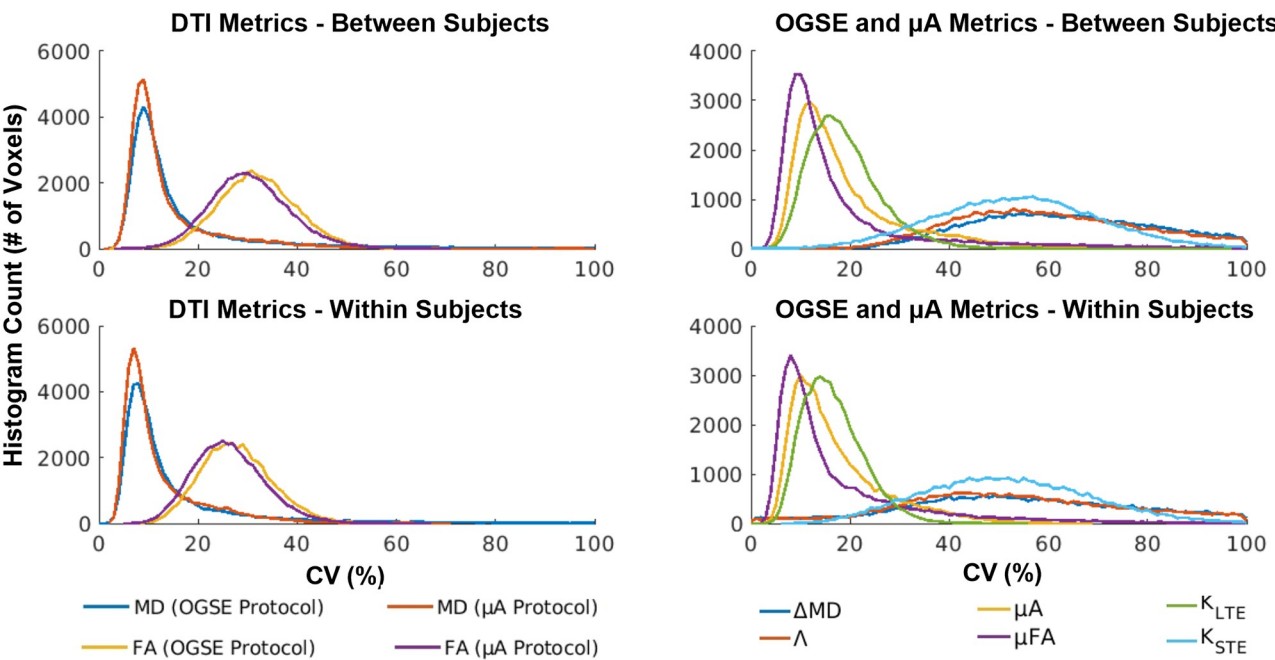

**Fig 11. Distribution of between and within subject whole brain voxel-wise CVs for the OGSE and µA metrics.** DTI metrics, acquired from both protocols, are shown as a reference.

ROI-based and voxel-wise analysis, indicating less variability within subjects on a test-retest basis.

## SNR analysis

Although the MP40 volume coil provides uniform whole-brain SNR (as shown in Fig 2), the surface coil may be preferred for applications focusing on certain regions of the brain, such as the cortex and the corpus callosum. Although higher CVs are observed farther away from the surface coil for all metrics (Fig 10), the gradient of SNR change observed for the surface coil (Fig 2) does not seem to affect the voxel-wise CV maps to the same extent, which could be due to the denoising quality. Furthermore, lowering the number of averages in the acquisition shows comparable ROI-based reproducibility (S1 Fig), which points to the robustness of the denoising and outlier removal in the pipeline. This also suggests that the reproducibility of the dMRI metrics is more heavily impacted by physiological effects (such as between-subject heterogeneity) and partial volume effects, compared to SNR.

## ROI-based reproducibility

Our ΔMD maps (Fig 4) show contrast which is consistent with recent observations in both *in vivo* and *ex vivo* OGSE studies in mouse brains by other groups [35, 71–73]. Aggarwal et al. related the higher OGSE contrast in the dentate gyrus layer of the hippocampal formation to densely packed neurons in the region [71], which simulations have indicated increase the rate of change in MD with frequency [74]. The very low values of ΔMD seen in certain regions of the gray matter are due to partial volume effects from CSF, as CSF exhibits negative values of ΔMD due to flow [12, 75]. ΔMD and Λ maps (Fig 4) show the same contrast, since the apparent diffusion dispersion rate is directly proportional to ΔMD. This relationship is also reflected

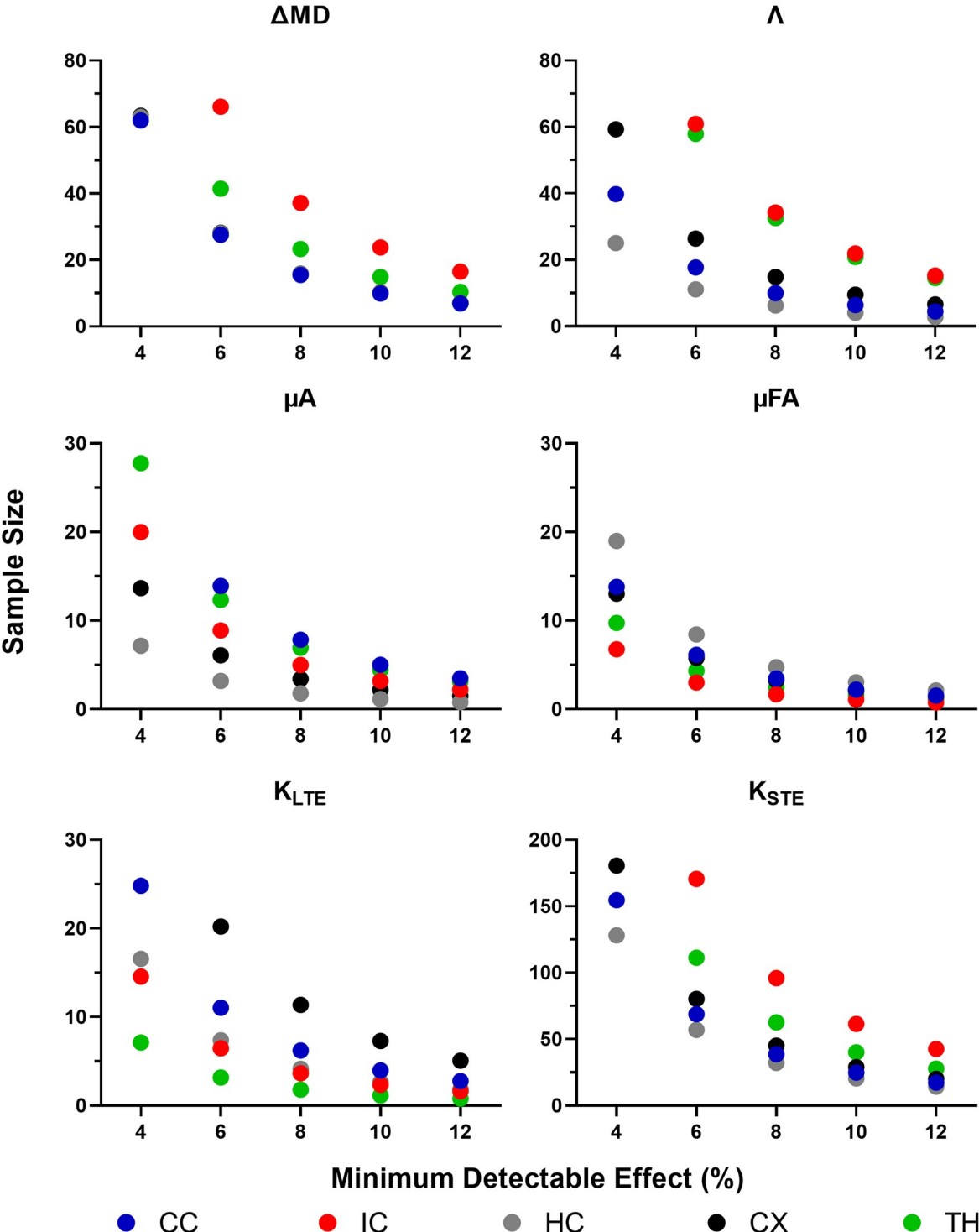

**Fig 12. Sample size estimation using a between-subjects approach.** Sample sizes required, calculated from ROI-based between-subject CVs, to detect a statistically significant effect within each ROI with a change in each metric of 4, 6, 8, 10, and 12%. Note that the sample size range varies between plots and sample sizes exceeding the range are not shown. ROIs are abbreviated as follows: CC—corpus callosum; IC —internal capsule; HC—hippocampus; CX—cortex; TH—thalamus.

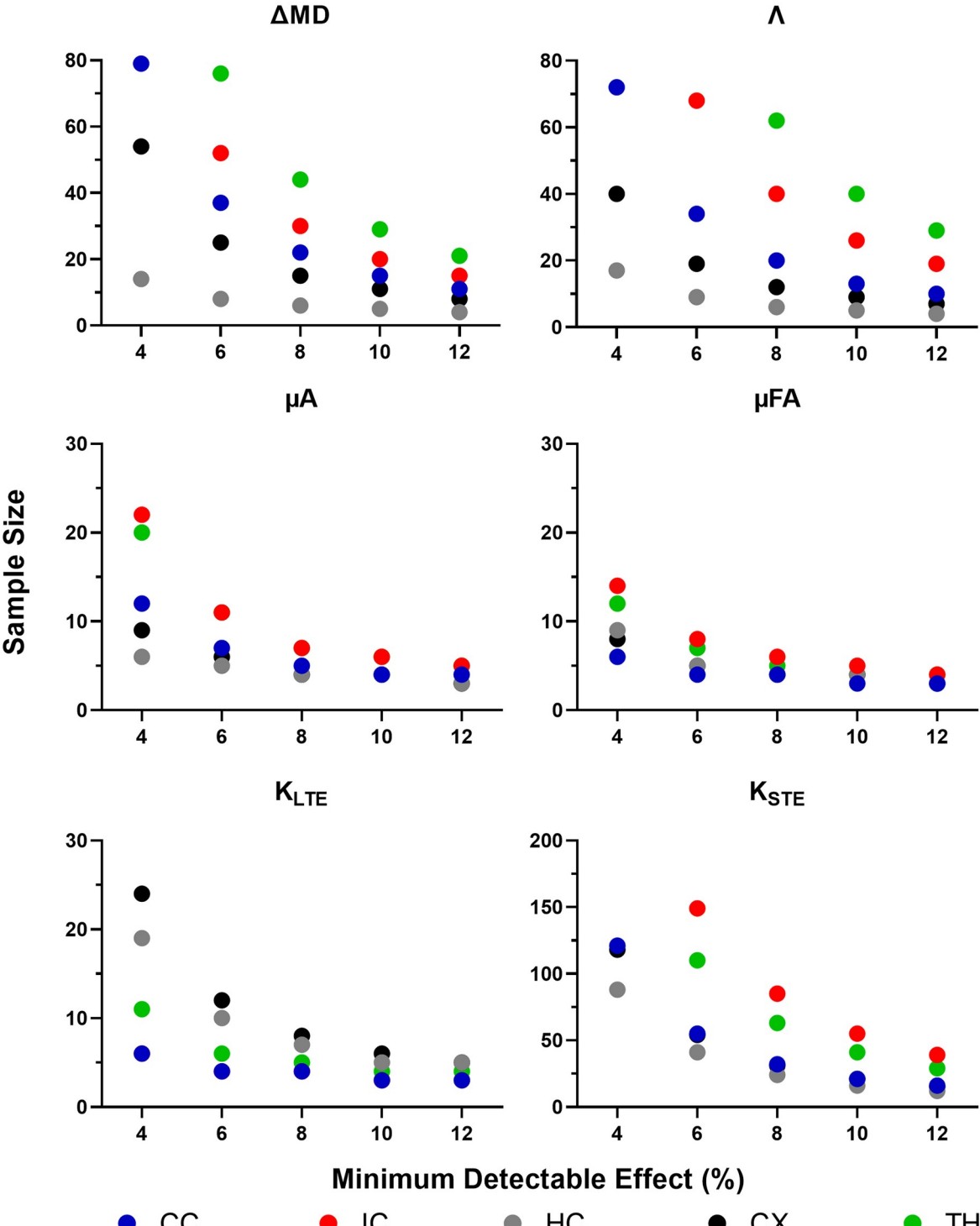

**Fig 13. Sample size estimation using a within-subjects approach.** Sample sizes required, calculated from the standard deviation of differences between test-retest mean values across subjects (assuming paired t-tests), to detect a statistically significant effect within each ROI with a change in each metric of 4, 6, 8, 10, and 12%. Note that the sample size range varies between plots and sample sizes exceeding the range are not shown. ROIs are abbreviated as follows: CC—corpus callosum; IC—internal capsule; HC—hippocampus; CX—cortex; TH—thalamus.

in the ΔMD and Λ ROI-based CVs (Fig 8) and voxel-wise CV maps (Fig 10), which are very similar. While ΔMD requires less scan time than Λ, as it requires only a single OGSE and PGSE acquisition, acquiring multiple frequencies allows probing of whether diseases may affect the power law scaling of MD with respect to frequency ($f^{0.5}$ was assumed here). Further, Λ is expected to be more robust in terms of reproducibility as it includes data from all OGSE acquisitions (as shown in Fig 5). This is reflected in our results by the smaller sample sizes needed to detect the same statistically significant changes in Λ, compared to ΔMD (Figs 12 and 13).

In the mean ROI analysis, the size and location of the ROIs influenced the reliability of the measurements. A greater distribution in the mean values for all metrics are observed in the internal capsule and thalamus (Fig 6), which are the smallest ROIs analyzed in this study. Similarly, higher CVs and a greater dispersion of CV values are observed in both smaller ROIs (Fig 8). This result leads to greater sample sizes being required to detect the same change in the smaller ROIs compared to the larger ROIs in some metrics (Fig 12). Thus, smaller ROIs lead to unreliable measurements due to less averaging and possibly a greater effect from slight registration inaccuracies. Both smaller ROIs are also positioned in the lower half of the brain, farther away from the surface coil. In addition to the location and the size of the ROIs, certain brain regions, such as the internal capsule, show higher between-subject anatomical variation [76], which would result in a higher between-subject CV. Furthermore, greater variability in terms of tissue microstructure, such as the greater variation of cell sizes and cell types in the thalamus [77], may also lead to higher CVs.

It is noteworthy that FA showed comparable reproducibility to Λ and μFA in the corpus callosum (i.e., white matter) and generally lower reproducibility in grey matter, which suggests sample sizes estimated using FA reproducibility would also be sufficient to investigate Λ and μFA. The lower reliability of FA in grey matter, compared to μFA, likely stems from its low value due to intravoxel dispersion of fiber orientations.

## Voxel-wise reproducibility

Voxel-wise analysis for specific ROIs (S2 Fig) shows that in general, the 3 ROIs shown (the corpus callosum, hippocampus, and cortex) follow the same trends. The corpus callosum shows a slightly lower CV peak than the gray matter regions for the more reproducible metrics (μA, μFA, and $K_{LTE}$). Overall, the within subject CV histograms have peaks at lower values than the between subject CV histograms, indicating less variability on a within subject test-retest basis. This is also noticeable in the between and within subject CV maps (Fig 10), with the within subject CV maps showing lower values overall.

One of the main reasons for the lack of reproducibility through voxel-wise analysis of ΔMD and Λ is likely CSF partial voluming. Since voxels with CSF can exhibit negative ΔMD and Λ values, whereas brain tissue shows positive ΔMD values, this leads to very high CVs (CVs > 60) in voxels impacted by CSF contamination, such as in regions with CSF in adjacent slices. This partial volume effect on ΔMD and Λ can be mitigated by using a higher resolution. However, this would also reduce SNR and longer scan times would be required to produce the same image quality. Voxel-wise analysis of ΔMD and Λ (from *in vivo* OGSE data) is not feasible given the resolution and scan time constraints. In contrast, ΔMD and Λ both show good reproducibility in the ROI analysis, where this partial volume effect is mitigated due to averaging. μA, μFA, and $K_{LTE}$ also show greater CVs in regions with CSF, such as the ventricles, arising from the very small values of these metrics in CSF.

As $K_{STE}$ values are intrinsically low in the brain [4, 41], higher CVs and greater dispersion of CV values are observed, even in the ROI analysis. Since $K_{STE}$ depends on the variance in

mean diffusivity, low $K_{STE}$ values point to a low variance in MD. This indicates similar sized cells across the brain, since a higher variance in cell size would lead to a higher variance in MD. In other words, the volume-weighted variance of cell size is low compared to the mean cell size. Unlike the other metrics explored in this study, $K_{STE}$ shows very low CVs in regions with CSF and in regions affected by CSF contamination (Fig 10), since $K_{STE}$ values are very high in CSF (Fig 6). As the CSF STE signal as a function of b-value decays very rapidly and reaches the noise floor, the fitting detects a false variance (very high $K_{STE}$) if high b-value data is not excluded [4]. The generally low reliability of $K_{STE}$ is likely due to a combination of its low value and the well-known sensitivity of kurtosis fitting to both physiological and thermal noise [78]. Notably, while ostensibly based on kurtosis fitting, μA and μFA do not suffer similar issues because no $2^{nd}$ order kurtosis fitting is required to estimate these metrics due to term cancellations that occur when the kurtosis difference between LTE and STE is evaluated to estimate these metrics [24]. Despite the low reliability, it is encouraging that the $K_{STE}$ maps (Fig 4) exhibit contrast which is comparable to $K_{STE}$ maps shown in a recent *in vivo* rodent study applying correlation tensor imaging (a DDE technique) [79].

## Sample size and minimum detectable effect

Given the current test-retest study design, small changes ($< 6\%$) can be detected in μA, μFA, and $K_{LTE}$, both between and within subjects, with moderate sample sizes of 10–15. With all minimum detectable changes explored (Figs 12 and 13), μFA was the most sensitive metric, followed by μA. ΔMD and Λ can detect moderate changes ($> 6\%$), given sample sizes of 12–15. $K_{STE}$ cannot detect small changes with sample sizes relevant to preclinical neuroimaging studies, unless compromises in scan time or resolution are made to improve SNR compared to the scans performed here.

## Limitations

It should be noted that the findings in this work are specific to the scan parameters used. Diffusion MRI is inherently a low SNR technique and high b-value acquisitions (from the μA protocol) and high oscillating gradient frequency acquisitions (from the OGSE protocol) result in even lower SNR. To acquire sufficient SNR, the voxel size was adjusted, with slice thickness set to 500 μm. Since our metrics are greatly impacted by partial volume effects (mostly from CSF), a higher resolution may provide more accurate and reproducible measurements. However, acquiring higher resolution with comparable SNR would require much greater scan time, which is not feasible for longitudinal *in vivo* neuroimaging studies, which are essential to characterize the progression of disease and injury recovery. Furthermore, a single channel transceive surface coil was used in this study and scan acceleration with parallel imaging was not possible. An option for obtaining more reliable ΔMD measures is to acquire only one PGSE and one OGSE scan, utilizing the same scan time of 45 minutes for the multifrequency OGSE protocol in this study. Thus, greater SNR and/or resolution can be achieved with more averaging. However, in doing so, one would lose the potential additional insight into microstructure organization and tissue integrity that multiple frequency analysis can provide if, for example, the $f^{0.5}$ power law scaling of MD changes in certain pathologies.

In the statistical analyses, it should be noted that for the within-subject calculation of CV, the standard deviation was determined from only two data points (the test and retest conditions). As a result, the standard deviation may not accurately represent the spread of data within the population, leading to an unknown bias in the resulting within-subject CV.

## Conclusion

In conclusion, we have investigated the reproducibility of OGSE and μA metrics in a rodent model at an ultra-high field strength. We have shown that the μA, μFA, and $K_{LTE}$ metrics (from the μA protocol) are reproducible in both ROI-based and voxel-wise analysis, while the ΔMD and Λ metrics (from the OGSE protocol) are only reproducible in ROI-based analysis. Given feasible sample sizes (10–15), μA, μFA, and $K_{LTE}$ may provide sensitivity to subtle microstructural changes (4–8%), while ΔMD and Λ may provide sensitivity to moderate changes ($> 6\%$). This work will provide insight into experiment design and sample size estimation for future longitudinal *in vivo* OGSE and μA microstructural dMRI studies at 9.4 T.

## Supporting information

**S1 Fig. ROI-based mean between subject and within subject coefficients of variation (CV) analysis for OGSE and μA metrics, acquired with fewer averages.** DTI metrics, MD and FA, acquired from both the OGSE and μA protocols, are shown as a reference. Values for the between subject condition represent the mean ± standard deviation over subjects (averaged over the test and retest timepoints). Values for the within subject condition represent the mean ± standard deviation between test and retest (averaged over the eight subjects). ROIs are abbreviated as follows: CC—corpus callosum; IC—internal capsule; HC—hippocampus; CX—cortex; TH—thalamus.
(TIF)

**S2 Fig. Distribution of voxel-wise between and within subject CVs within each ROI.**
(TIF)

## Author Contributions

**Conceptualization:** Naila Rahman, Arthur Brown, Corey A. Baron.

**Data curation:** Naila Rahman, Kathy Xu, Corey A. Baron.

**Formal analysis:** Naila Rahman, Mohammad Omer.

**Funding acquisition:** Corey A. Baron.

**Investigation:** Naila Rahman.

**Methodology:** Naila Rahman, Kathy Xu, Arthur Brown, Corey A. Baron.

**Project administration:** Naila Rahman.

**Resources:** Kathy Xu, Matthew D. Budde, Arthur Brown, Corey A. Baron.

**Software:** Naila Rahman, Matthew D. Budde, Corey A. Baron.

**Supervision:** Arthur Brown, Corey A. Baron.

**Validation:** Naila Rahman, Corey A. Baron.

**Visualization:** Naila Rahman.

**Writing – original draft:** Naila Rahman.

**Writing – review & editing:** Naila Rahman, Kathy Xu, Mohammad Omer, Matthew D. Budde, Arthur Brown, Corey A. Baron.

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
