## [Decision Letter · Decision Letter 0]

31 Aug 2021

PONE-D-21-23637

Test-retest reproducibility of in vivo oscillating gradient and microscopic anisotropy diffusion MRI in mice at 9.4 Tesla

PLOS ONE

Dear Dr. Rahman,

Thank you for submitting your manuscript to PLOS ONE. After careful consideration, we feel that it has merit but does not fully meet PLOS ONE’s publication criteria as it currently stands. Therefore, we invite you to submit a revised version of the manuscript that addresses the points raised during the review process.

Both Reviewers have proposed a number of changes, which are complementary to one another. These suggestions are in line with my own reading of the manuscript. As such, I recommend that you follow their suggestions as best as possible. I recognize that scanning additional mice, as suggested by Reviewer 2, might not be feasible, but please consider the alternative option proposed. 

We look forward to receiving your revised manuscript.

Kind regards,

Niels Bergsland

Academic Editor

PLOS ONE

Journal Requirements:

2. Please amend your Methods section if the mice were sacrificed at the end of the study. If so, please specify the method of euthanasia.

Reviewers' comments:

Reviewer's Responses to Questions

**Comments to the Author**

1. Is the manuscript technically sound, and do the data support the conclusions?

Reviewer #1: Yes

Reviewer #2: Yes

2. Has the statistical analysis been performed appropriately and rigorously? 

Reviewer #1: Yes

Reviewer #2: Yes

3. Have the authors made all data underlying the findings in their manuscript fully available?

Reviewer #1: Yes

Reviewer #2: Yes

4. Is the manuscript presented in an intelligible fashion and written in standard English?

Reviewer #1: Yes

Reviewer #2: Yes

5. Review Comments to the Author

Reviewer #1: The manuscript describes a test-retest analysis of diffusion MRI metrics derived from recently proposed technique on a 9.4T pre-clinical system. The manuscript is overall well written and the results are well presented and of interest to the neuroimaging community, nevertheless, several issues need to be addressed / corrected:

Major points:

1. The authors should add an SNR analysis of their data. The results and their interpretation depend on the actual noise level, that, at the moment is included only in the discussion. Since the different repetitions are acquired separately, maybe the authors could consider the effect of different number of averages (and SNR) on the results. Also, this study employs an in-house built coil. How does the SNR compare to other standard coils? How do the results translate to other equipment?

2. Introduction – overall the authors should make clearer the fact that they are investigating two distinct methodologies, which are not combined in any way in this work. Also, the mentioning of microscopic (intra-compartment) kurtosis in the introduction is not needed, as it is not used in this work and confusing. The authors can add this to the discussion section instead. (If uK is discussed, then the authors should also include a reference to the recent work on stroke https://www.biorxiv.org/content/10.1101/2021.02.20.432088v1.full)

3. In the variability analysis, the authors should also include standard DTI metrics (MD, FA) as a reference.

4. In the methods, the authors should provide more details about the computation of KLTE, KSTE, uA and uFA and provide the expressions used to calculate the metrics, as there is some confusion with the notation and meaning of the different parameters. The authors state “anisotropic kurtosis (KLTE – arising from the LTE acquisitions)” – However, by fitting a kurtosis term to the LTE acquisition alone (either to the entire data or the powder average), one obtains the total kurtosis, and not just the anisotropic part. So, the derivation and meaning of parameters need to be clarified.

5. OGSE sequences and Figure 1- the gradient amplitudes in the plots appear to be the same, while in the text, it’s specified that the b-value is the same, leading to different gradient strengths. Also, the gradient duration appears different for different frequencies, while in the text it is the same. Please make sure the plots and the text are accurate. Also, the shape of the sequence at 50 Hz is different than the others. How does the power spectrum look? Maybe add the power spectra for all sequences to Figure 1. For the sequence at 0 Hz, please specify it’s a PGSE with a certain gradient duration and diffusion time, rather than OGSE with 0Hz. Also, for sequences in Figure 1A and 1B, where the gradients are not refocused before and after the 180 pulse, did the authors consider the effect of the imaging gradients in the calculation of the b-value?

Minor points:

Line 95 – diffusion times achievable in PGSE can probe displacements on the order of 10-30 um, please rephrase.

In the introduction, please specify that OGSE with low frequencies have also been shown to provide better contrast to cylinder diameter in the presence of orientation dispersion (Drobnjak et al, MRM 2015, Nilsson et al, NMR Biomed)

Line 122-123 - DDE doesn’t necessarily require longer echo times compared to STE to achieve the same b-value. Please rephrase.

The earlier DDE studies (Mitra 95, Cheng and Cory 99, Shemesh and Cohen 2011, Jespersen et al NMR Biomed 2013, etc) should also be cited.

Notation – KLTE and KSTE, the LTE and STE should be with subscripts.

Line 138 – Please also add citations to the following studies when referring to the use of OGSE for mapping axon diameter (https://pubmed.ncbi.nlm.nih.gov/25809657/, https://pubmed.ncbi.nlm.nih.gov/28774648/, https://discovery.ucl.ac.uk/id/eprint/1396458/ )

Line 139 – please provide a more complete citation list relating uA to pore shape (Mitra 95, Cheng and Cory 99, Shemesh and Cohen 2011, Ozarslan JMR 2009, Lasic et al Front Phys 2013, Jespersen et al NMR Biomed 2013, Ianus et al NMR Biomed 2016, etc)

Line 161 - typo – remove ‘an’

Line 169 - what was the age of the animals?

How were the slices placed for the test and retest? Did the authors ensure their consistent positioning? Please specify.

How was EDDY correction applied and did it have any effect? As per the documentation it requires images with close to opposite gradient directions, which I don’t believe it is the case for 4 or 6 directions.

Line 270 ROI analysis – why for OGSE the signal was averaged in the ROI first, and then MD was fitted, while uA was fitted voxelwise? The analysis should be done in a similar way.

Line 277 How was voxelwise analysis done for uA?

Figure 2 – when plotting the diffusion images / maps, please make sure the image x and y dimensions follow the voxel sizes, as the brain appears a bit squished due to the anisotropic in plane resolution.

Line 289 – Kiso should be high in regions of partial volume with CSF, rather than just CSF, which is likely the case here due to the thickness of the slices. Please correct.

Line 307-308 – is the variability just related to the size of the ROI? The thalamus is not a very small ROI, but shows more variability in terms of tissue microstructure, as can be seen in an atlas or from stains showing myelin content for example.

Line 339 – estimated bias – estimated mean? The authors just stated the biases are negligible.

The Discussion should have a clear subtitle related to limitations.

Reviewer #2: This contribution by Rahman et al. (PONE-D-21-23637) provides a thorough characterisation of inter- and intra-subject variability of metrics derived from advanced dMRI approaches including spherical tensor encoding and OGSE MRI in preclinical imaging. MRI experiments were performed on N=8 mice using a preclinical scanner (9.4T) and state-of-the-art gradients (1T/m) for the OGSE experiments. The mice were scanned twice within a span of five days, to allow for intra-subject variability assessments. Typical scan times, acquisitions, and analysis pipelines were used to make the findings as generalizable as possible. Inter- and intra-animal coefficients of variation and Bland-Altman metrics were computed, and the authors report a high voxel-wise reproducibility for most metrics except for dispersion rates, mean diffusivity subtractions, and K_STE, which were only reproducible with ROI based analyses.

Overall, I find the study to be important to the field as it highlights constraints that need to be accounted for in many future studies. The study was well performed and will certainly fit the PLOS ONE readership, and the results are sufficiently novel for publication.

I have only a few relatively minor comments which I am sure the authors can address upon revision:

1. I may have missed the point, but in line 170 the authors state that they chose the sample size to “...reflect common practices...”. While I do understand this, I would assume that to estimate the true reproducibility of the metrics, the authors would have to choose the cohort based on effect sizes and not on the “common practice”.

2. Outliers: I am wondering whether the authors tested for outliers and removed them, both voxelwise and animal-wise? This could dramatically change the conclusions.

3. I am suspecting (maybe I am wrong, but I have strong reasons to say this...) that the poorer CV’s in the OGSE data is not an inherent property, but is mainly due to the quality of denoising. Bruker EPI data is “spitted out” with zero-filling, which creates spatial correlations that significantly reduce the denoising quality (just iFT the Bruker b=0 and you will see the effect). Partial Fourier acquisitions exacerbate these further. I’d suggest the authors to either:

a- acknowledge these limitations explicitly in the paper and/or

b- acquire a couple of more animals without partial Fourier, denoise the data after removing the zero-filling (which is due to the gridding process in the recon) and see whether the CV’s improve.

4. In the Introduction, when discussing uA from LTE and STE, the authors should state clearly that the multiple gaussian component assumption was made, which may not be appropriate in many cases (there is a discussion in the end but I think it is appropriate to highlight this clearly in the Intro).

5. I think that in a study like this, it is appropriate to show raw data and preprocessed (denoised, unrung, etc.) data before the maps (even if in SI, though I always find this to be very important and try myself to keep it within the main text).

6. In addition, I believe it will be useful to show explicit S/N maps for the acquisitions at the zero and highest b-values, respectively.

Noam Shemesh

6. PLOS authors have the option to publish the peer review history of their article (what does this mean?). If published, this will include your full peer review and any attached files.

Reviewer #1: No

Reviewer #2: **Yes: **Noam Shemesh

---

## [Author Response · Author response to Decision Letter 0]

13 Oct 2021

We would like to thank the reviewers for their insightful comments and suggestions, which have improved the quality of the manuscript. Our responses to the suggestions can be found below.

Miscellaneous changes:

We have increased the sample size from 8 to 12 due to continued data acquisition during the review of the first submission. Notably, our findings have remained similar to our first submission.

Journal Requirements:

Please amend your Methods section if the mice were sacrificed at the end of the study. If so, please specify the method of euthanasia.

We have included this in the Methods (under the Subjects heading).

Reviewers’ Comments

Reviewer #1: The manuscript describes a test-retest analysis of diffusion MRI metrics derived from recently proposed technique on a 9.4T pre-clinical system. The manuscript is overall well written and the results are well presented and of interest to the neuroimaging community, nevertheless, several issues need to be addressed / corrected:

Major points:

1. The authors should add an SNR analysis of their data. The results and their interpretation depend on the actual noise level, that, at the moment is included only in the discussion. Since the different repetitions are acquired separately, maybe the authors could consider the effect of different number of averages (and SNR) on the results. Also, this study employs an in-house built coil. How does the SNR compare to other standard coils? How do the results translate to other equipment?

Thank you for suggesting this important addition to the analysis. SNR maps are now shown with fewer averages acquired and using an MP40 volume coil (Figure 2). ROI-based CV analysis was also performed using fewer averages (Supplemental Figure 1).

2. Introduction – overall the authors should make clearer the fact that they are investigating two distinct methodologies, which are not combined in any way in this work. Also, the mentioning of microscopic (intra-compartment) kurtosis in the introduction is not needed, as it is not used in this work and confusing. The authors can add this to the discussion section instead. (If uK is discussed, then the authors should also include a reference to the recent work on stroke https://www.biorxiv.org/content/10.1101/2021.02.20.432088v1.full)

Thank you – this makes the intro clearer. We have stated that we are evaluating two distinct dMRI methods separately and the microscopic kurtosis reference has been removed.

3. In the variability analysis, the authors should also include standard DTI metrics (MD, FA) as a reference.

Thank you for this suggestion. We have now included standard DTI metrics as a reference in both the ROI-based (Fig. 8) and voxel-wise (Fig. 10) variability analysis. The MD and FA acquired from both OGSE and µA protocols were both included via the PGSE (b800) and LTE (using the b1000 shell) acquisitions, respectively. FA showed a relatively low reproducibility in the gray matter compared to the OGSE and µA metrics, which we now briefly address in the Discussion.

4. In the methods, the authors should provide more details about the computation of KLTE, KSTE, uA and uFA and provide the expressions used to calculate the metrics, as there is some confusion with the notation and meaning of the different parameters. The authors state “anisotropic kurtosis (KLTE – arising from the LTE acquisitions)” – However, by fitting a kurtosis term to the LTE acquisition alone (either to the entire data or the powder average), one obtains the total kurtosis, and not just the anisotropic part. So, the derivation and meaning of parameters need to be clarified.

Our apologies – this has been corrected in the Introduction and a more detailed explanation of the meaning of the parameters from the µA protocol has been included, with more computation details in the Methods.

5. OGSE sequences and Figure 1- the gradient amplitudes in the plots appear to be the same, while in the text, it’s specified that the b-value is the same, leading to different gradient strengths. Also, the gradient duration appears different for different frequencies, while in the text it is the same. Please make sure the plots and the text are accurate. Also, the shape of the sequence at 50 Hz is different than the others. How does the power spectrum look? Maybe add the power spectra for all sequences to Figure 1. For the sequence at 0 Hz, please specify it’s a PGSE with a certain gradient duration and diffusion time, rather than OGSE with 0Hz. Also, for sequences in Figure 1A and 1B, where the gradients are not refocused before and after the 180 pulse, did the authors consider the effect of the imaging gradients in the calculation of the b-value?

The gradient amplitudes in Figure 1 have been corrected. Our apologies – the gradient duration is actually different for the different frequencies and this has been explained in the Methods and in Figure 1 (note that the TE remains consistent). Power spectra have now been added for the sequences in Figure 1, which makes it clear the power spectra for the 50 Hz scan is comparable to typical OGSE encodings. The 50 Hz sequence is based on a waveform recently designed in our lab (manuscript under review with MRM; in this manuscript we have referenced the preprint), which we now describe in the Methods section (under the In vivo MRI – OGSE dMRI subheading). The sequence implementation automatically takes into account the effect of the imaging gradients in the calculation of all b-values, which includes the sequences in Figure 1A and 1B.

Minor points:

Line 95 – diffusion times achievable in PGSE can probe displacements on the order of 10-30 um, please rephrase.

We’ve rephrased this.

In the introduction, please specify that OGSE with low frequencies have also been shown to provide better contrast to cylinder diameter in the presence of orientation dispersion (Drobnjak et al, MRM 2015, Nilsson et al, NMR Biomed)

Thank you! We’ve included this.

Line 122-123 - DDE doesn’t necessarily require longer echo times compared to STE to achieve the same b-value. Please rephrase.

Thank you! We’ve rephrased this.

The earlier DDE studies (Mitra 95, Cheng and Cory 99, Shemesh and Cohen 2011, Jespersen et al NMR Biomed 2013, etc) should also be cited.

Notation – KLTE and KSTE, the LTE and STE should be with subscripts.

The above citations have been included.

Line 138 – Please also add citations to the following studies when referring to the use of OGSE for mapping axon diameter (https://pubmed.ncbi.nlm.nih.gov/25809657/, https://pubmed.ncbi.nlm.nih.gov/28774648/, https://discovery.ucl.ac.uk/id/eprint/1396458/ )

Thank you for these interesting reads. We’ve included the studies above.

Line 139 – please provide a more complete citation list relating uA to pore shape (Mitra 95, Cheng and Cory 99, Shemesh and Cohen 2011, Ozarslan JMR 2009, Lasic et al Front Phys 2013, Jespersen et al NMR Biomed 2013, Ianus et al NMR Biomed 2016, etc)

The above citations have been included.

Line 161 - typo – remove ‘an’

Removed.

Line 169 - what was the age of the animals?

How were the slices placed for the test and retest? Did the authors ensure their consistent positioning? Please specify.

The age and positioning of the mouse head has been addressed in the Methods (under the Subjects and In vivo MRI headings respectively).

How was EDDY correction applied and did it have any effect? As per the documentation it requires images with close to opposite gradient directions, which I don’t believe it is the case for 4 or 6 directions.

According to the documentation for FSL EDDY usage, a set of diffusion encoding directions that span the entire sphere and not just a half-sphere or a blip-up-blip-down (phase encode reversed) acquisition is beneficial but not required. For eddy to work well there also needs to be a minimum number of diffusion directions, and as per the documentation, the minimum is ~10-15 directions for a b-value of 1500. 

In our experience, EDDY corrects eddy-current induced distortions well even with fewer directions, such as our 10-direction scheme. Example FA maps before and after topup/eddy are applied are shown below. Incorrect FA values in the superior region can be seen in part A and the axial image looks more stretched out along the PE direction.

A) FA Map generated before topup/eddy

B) FA Map generated after topup/eddy

For the µA protocol, we used 12 and 30 directions for b-values of 1000 and 2000, respectively. The directions were generated using the ‘dirgen’ command from MRtrix 3.0. For OGSE, we generated 10 directions, distributed over half a cube, as shown below:

The 10 directions don’t actually “combine” the 4 or 6 direction schemes, as we had previously stated. Sorry about the confusion! That has been removed from the Methods. Additionally, EDDY seems to work well even for a 4-direction scheme, such as the 4-direction OGSE performed by us in earlier work (Arbabi et al., MRM, 2020).

Line 270 ROI analysis – why for OGSE the signal was averaged in the ROI first, and then MD was fitted, while uA was fitted voxelwise? The analysis should be done in a similar way.

Sorry, we realize this should have been explained in a more coherent manner! We’ve tried to make this clearer in the Methods – ROI Analysis section. We averaged MD for each OGSE frequency, then ΔMD was calculated and Λ was fitted. The µA metrics are computed directly from the signal, but ΔMD and Λ are computed from MD and not directly from the signal. So, for both OGSE and µA, averaging for each ROI is performed over the first non-signal parameter computed.

Line 277 How was voxelwise analysis done for uA?

The same parameter maps (from the µA protocol) were used for ROI-based and voxelwise analyses. The voxelwise analysis is done by calculating voxelwise CVs, instead of ROI-based CVs.

Figure 2 – when plotting the diffusion images / maps, please make sure the image x and y dimensions follow the voxel sizes, as the brain appears a bit squished due to the anisotropic in plane resolution.

We’ve made changes to Figure 4.

Line 289 – Kiso should be high in regions of partial volume with CSF, rather than just CSF, which is likely the case here due to the thickness of the slices. Please correct.

This has been corrected.

Line 307-308 – is the variability just related to the size of the ROI? The thalamus is not a very small ROI, but shows more variability in terms of tissue microstructure, as can be seen in an atlas or from stains showing myelin content for example.

Thank you for this insight. We’ve mentioned this in the Discussion (under the ROI-based Reproducibility heading).

Line 339 – estimated bias – estimated mean? The authors just stated the biases are negligible.

This has been rephrased.

The Discussion should have a clear subtitle related to limitations.

Subtitles have been added to the Discussion.

Reviewer #2: This contribution by Rahman et al. (PONE-D-21-23637) provides a thorough characterisation of inter- and intra-subject variability of metrics derived from advanced dMRI approaches including spherical tensor encoding and OGSE MRI in preclinical imaging. MRI experiments were performed on N=8 mice using a preclinical scanner (9.4T) and state-of-the-art gradients (1T/m) for the OGSE experiments. The mice were scanned twice within a span of five days, to allow for intra-subject variability assessments. Typical scan times, acquisitions, and analysis pipelines were used to make the findings as generalizable as possible. Inter- and intra-animal coefficients of variation and Bland-Altman metrics were computed, and the authors report a high voxel-wise reproducibility for most metrics except for dispersion rates, mean diffusivity subtractions, and K_STE, which were only reproducible with ROI based analyses.

Overall, I find the study to be important to the field as it highlights constraints that need to be accounted for in many future studies. The study was well performed and will certainly fit the PLOS ONE readership, and the results are sufficiently novel for publication.

I have only a few relatively minor comments which I am sure the authors can address upon revision:

1. I may have missed the point, but in line 170 the authors state that they chose the sample size to “...reflect common practices...”. While I do understand this, I would assume that to estimate the true reproducibility of the metrics, the authors would have to choose the cohort based on effect sizes and not on the “common practice”.

In retrospect, we realize this wording may have been confusing. We’ve changed the wording to more correctly reflect how we chose our sample size, simply based on similar sample sizes used in other preclinical imaging studies. We completely agree that the best practice would be to choose the cohort based on effect sizes. We are aware of how effect sizes can be used to calculate sample size, if we are looking for differences between groups. However, we are not aware of any method that allows us to use effect size to estimate variance accurately or of other similar reproducibility studies using effect size to estimate sample size. This has intrigued us as well!

2. Outliers: I am wondering whether the authors tested for outliers and removed them, both voxelwise and animal-wise? This could dramatically change the conclusions.

Thank you for this suggestion! We’ve added outlier detection to the pipeline. This has overall slightly improved reliability, but trends and conclusions have not dramatically changed.

3. I am suspecting (maybe I am wrong, but I have strong reasons to say this...) that the poorer CV’s in the OGSE data is not an inherent property, but is mainly due to the quality of denoising. Bruker EPI data is “spitted out” with zero-filling, which creates spatial correlations that significantly reduce the denoising quality (just iFT the Bruker b=0 and you will see the effect). Partial Fourier acquisitions exacerbate these further. I’d suggest the authors to either:

a- acknowledge these limitations explicitly in the paper and/or

b- acquire a couple of more animals without partial Fourier, denoise the data after removing the zero-filling (which is due to the gridding process in the recon) and see whether the CV’s improve.

We happened to also be aware of this subtle Bruker limitation, and retrospectively filled in the zero-filled portion data using POCS or “Projection onto Convex Sets” (Haacke, Lindskog, and Lin, JMR, 1991). This has now been included in the Methods section.

4. In the Introduction, when discussing uA from LTE and STE, the authors should state clearly that the multiple gaussian component assumption was made, which may not be appropriate in many cases (there is a discussion in the end but I think it is appropriate to highlight this clearly in the Intro).

We’ve included this in the Intro.

5. I think that in a study like this, it is appropriate to show raw data and preprocessed (denoised, unrung, etc.) data before the maps (even if in SI, though I always find this to be very important and try myself to keep it within the main text).

Raw and preprocessed data has been included in Figure 3.

6. In addition, I believe it will be useful to show explicit S/N maps for the acquisitions at the zero and highest b-values, respectively.

Thank you for this suggestion! SNR maps are now included in Figure 2. We’ve also shown SNR maps using fewer averages and using the MP40 volume coil.

---

## [Decision Letter · Decision Letter 1]

25 Oct 2021

Test-retest reproducibility of in vivo oscillating gradient and microscopic anisotropy diffusion MRI in mice at 9.4 Tesla

PONE-D-21-23637R1

Dear Dr. Rahman,

We’re pleased to inform you that your manuscript has been judged scientifically suitable for publication and will be formally accepted for publication once it meets all outstanding technical requirements.

Kind regards,

Niels Bergsland

Academic Editor

PLOS ONE

Additional Editor Comments (optional):

Reviewers' comments:

Reviewer's Responses to Questions

**Comments to the Author**

1. If the authors have adequately addressed your comments raised in a previous round of review and you feel that this manuscript is now acceptable for publication, you may indicate that here to bypass the “Comments to the Author” section, enter your conflict of interest statement in the “Confidential to Editor” section, and submit your "Accept" recommendation.

Reviewer #1: All comments have been addressed

Reviewer #2: All comments have been addressed

2. Is the manuscript technically sound, and do the data support the conclusions?

Reviewer #1: Yes

Reviewer #2: Yes

3. Has the statistical analysis been performed appropriately and rigorously? 

Reviewer #1: Yes

Reviewer #2: Yes

4. Have the authors made all data underlying the findings in their manuscript fully available?

Reviewer #1: Yes

Reviewer #2: Yes

5. Is the manuscript presented in an intelligible fashion and written in standard English?

Reviewer #1: Yes

Reviewer #2: Yes

6. Review Comments to the Author

Reviewer #1: The authors have addressed all my comments and added the necessary information! Congratulations for a very nice work.

It would be very nice if the authors would also make the implementation of the sequences available to the pre-clinical MRI community and if they could add a sentence in that regard.

Reviewer #2: (No Response)

7. PLOS authors have the option to publish the peer review history of their article (what does this mean?). If published, this will include your full peer review and any attached files.

Reviewer #1: **Yes: **Andrada Ianus

Reviewer #2: No

---

## [Editor Report · Acceptance letter]

29 Oct 2021

PONE-D-21-23637R1 

Test-retest reproducibility of *in vivo* oscillating gradient and microscopic anisotropy diffusion MRI in mice at 9.4 Tesla 

Dear Dr. Rahman:

I'm pleased to inform you that your manuscript has been deemed suitable for publication in PLOS ONE. Congratulations! Your manuscript is now with our production department. 

Kind regards, 

on behalf of

Dr. Niels Bergsland 

Academic Editor

PLOS ONE